# Discovery of van Hove singularities: electronic fingerprints of 3Q magnetic order in a van der Waals quantum magnet

Hai-Lan Luo [1,2], Josue Rodriguez [1], Debasis Dutta[3], Maximilian Huber[2], Haoyue Jiang[2,4], Luca Moreschini[1,2], Catherine Xu [1], Alexei Fedorov [5], Chris Jozwiak [5], Aaron Bostwick [5], Guoqing Chang [3], James G. Analytis [1,6,7], Dung-Hai Lee [1,2] & Alessandra Lanzara [1,2,7] ✉

Magnetically intercalated transition metal dichalcogenides are emerging as a rich platform for exploring exotic quantum states in van der Waals magnets. Among them, $Co_xTaS_2$ has attracted intense interest following the recent discovery of a distinctive **3Q** magnetic ground state and a pronounced topological Hall effect below a critical doping of $x \approx 1/3$, both intimately tied to cobalt concentration. To date, direct signatures of this enigmatic **3Q** magnetic order in the electronic structure remain elusive. Here we report a comprehensive doping dependent angle resolved photoemission spectroscopy study that unveils these long-sought fingerprints. Our data reveal an unexpected inverse-Mexican-hat dispersion along the K-M-K′ direction, accompanied by two van Hove singularities. These features are consistent with theoretical predictions for a **3Q** magnetic order near three-quarters band filling on a cobalt triangular lattice. These results provide evidence of **3Q** magnetic order in the electronic structure, establishing TMD van der Waals magnets as tunable materials to explore the interplay between magnetism and topology.

The family of transition metal dichalcogenides (TMDs) $MX_2$ (*M*: transition-metal atoms, *X*: chalcogen atoms) host layered structures with van der Waals (vdW) gaps and exhibits a plethora of exotic properties, such as superconductivity[1], charge density waves (CDW)[2], and topological electronic states[3]. Magnetically intercalated TMDs, as an important class of van der Waals magnets, open a new paradigm in TMD research and provide a versatile platform to investigate the interplay between magnetic order and electronic properties. This newly discovered variant of TMDs is formed by inserting magnetic ions into the vdW gaps of 2*H*-TMDs. The intercalated magnetic layers can induce magnetic moments that couple with the host conducting layers, giving rise to a wide class of interesting magnetic orders and

electronic properties. One class of these new materials is $TM_{1/3}(Ta, Nb)S_2$ (TM: 3d transition-metal atoms), where the TM ions form a triangular lattices with a $\sqrt{3} \times \sqrt{3}$ superlattice with respect to the unit cell of the host compounds $(Ta, Nb)S_2$. Depending on the host TMD layers and intercalants, highly tunable magnetic orders and interesting physical phenomena have been reported[4]. For example, V or Ni intercalation leads to a ferromagnetic (FM) order within the V or Ni plane and an antiferromagnetic (AFM) order along the out-of-plane direction[5–7]; the intercalation of Cr or Mn leads to a predominant FM order[8], characterized by a chiral helimagnetic structure in $(Cr, Mn)_{1/3}NbS_2$[8–10]; $Fe_{1/3}NbS_2$[11–13] shows AFM order with the magnetic moments predominantly aligned along the *c*-axis, while $Fe_{1/3}TaS_2$

[1]Department of Physics, University of California, Berkeley, Berkeley, CA, USA. [2]Materials Sciences Division, Lawrence Berkeley National Laboratory, Berkeley, CA, USA. [3]Division of Physics and Applied Physics, School of Physical and Mathematical Sciences, Nanyang Technological University, Singapore, Singapore. [4]Graduate Group in Applied Science and Technology, University of California, Berkeley, Berkeley, CA, USA. [5]Advanced Light Source, Lawrence Berkeley National Laboratory, Berkeley, CA, USA. [6]CIFAR Quantum Materials, CIFAR, Toronto, ON, Canada. [7]Kavli Energy NanoScience Institute, University of California, Berkeley, Berkeley, CA, USA. ✉e-mail: alanzara@lbl.gov

shows FM order along the $c$-axis[14]; in the case of Co intercalation, $Co_{1/3}NbS_2$ shows collinear in-plane AFM order and weak $c$-axis FM order[15,16]. Moreover, it is observed that slight changes in the stoichiometry of the intercalated ions can modify the magnetic order and transport properties. For example, in $Co_xNbS_2$, the Co content can tune the magnitude of the anomalous Hall effect in the proximity of $x = 1/3$[17].

The report of the unusual magnetic order with three non-coplanar ordering wavevector (the 3$\mathbf{Q}$ antiferromagnetic order)[18,19] coexisting with a pronounced doping-dependent topological Hall effect in $Co_xTaS_2$[20] has spurred a lot of interest in the study of the Co-intercalated TaS$_2$ family. It is believed that the topological Hall effect originates from the real-space Berry curvature induced by the scalar spin chirality associated with a non-coplanar magnetic order[18]. Importantly, slight variations in cobalt doping across the critical point $x = 1/3$ have dramatic effects on the magnetic order and the topological Hall conductance of $Co_xTaS_2$[20]. As of today, very little is known of the fingerprints of this peculiar magnetic order on the electronic structure of this material, and more importantly, on the potential topological nature of such order.

Here, we present a comprehensive angle-resolved photoemission spectroscopy (ARPES) study of the Co doping-dependent electronic structure of $Co_xTaS_2$ in the range of $x = 0.29$-$0.36$ across the transition from 3$\mathbf{Q}$ to helical AFM order and compare the results with the pristine 2$H$-TaS$_2$. The data reveal that, in addition to electron-doping the TaS$_2$-derived bands, the Co intercalation induces new shallow bands originating from Co-orbitals. Furthermore, within the studied $x$ range, we find the Co intercalants mainly dope the TaS$_2$-derived bands and have little effect on the Co-derived bands. Combined with potassium deposition experiments, we identify a high density of states in the Co-derived bands. Calculations on a triangular lattice, both without and with 3$\mathbf{Q}$ order, reveal that the Co-derived bands with 3/4-filling host van Hove singularities with divergent density of states near the Fermi level, consistent with the doping-dependent results. Furthermore, the Fermi surface and band structure are modified by the presence of the 3$\mathbf{Q}$ order, with corresponding signatures observed in our data.

## Results and discussion
### Structure and physical property of $Co_{1/3}TaS_2$
The crystal symmetry of $Co_{1/3}TaS_2$ is the non-centrosymmetric space group $P6_322$ (No. 182) sketched in Fig. 1a. Magnetic Co atoms are intercalated into the van der Waals gap in the layered compound 2$H$-TaS$_2$. The Co intercalants form a $\sqrt{3} \times \sqrt{3}$ triangular lattice rotated by 30° with respect to the $1 \times 1$ unit cell of TaS$_2$ (Fig. 1b). The top view of the crystal structure in Fig. 1b indicates that Co lies underneath or above the Ta atoms occupying the $2c$ Wyckoff positions. In the nearly-stoichiometric samples with $x = 1/3$, the $2c$ sites are fully occupied; for lower/higher doping, vacancies at the $2c$/occupancy at the $2b$ sites are observed[21]. Experimentally, tuning the Co doping across $x = 1/3$ can drastically change the bulk properties of $Co_xTaS_2$. Specifically, as shown in Fig. 1c, $Co_xTaS_2$ with $x \leq 0.325$ exhibits two magnetic phases, single-$\mathbf{Q}$ AFM order at higher T and triple-$\mathbf{Q}$ (3$\mathbf{Q}$) AFM order at lower T (see 3$\mathbf{Q}$ magnetic order in Fig. 1d), where the latter coexists with a pronounced topological Hall effect (Topo-HE)[19]. For $x > 0.325$, a coplanar helical AFM order with zero Hall conductivity emerges[20]. From here onward, we will refer to $x = 0.33$ as the critical doping $x_c$.

### Comparative analysis of the electronic structures of undoped and Cobalt-doped 2$H$-TaS$_2$
Figure 1e, f compare the low-temperature (7 K) Fermi surfaces of the undoped compound 2$H$-TaS$_2$ with that of $Co_{0.32}TaS_2$. The experimentally extracted Fermi surface sheets are plotted in the upper-left quadrants. For 2$H$-TaS$_2$ (Fig. 1e), the Fermi surfaces consist of two circular hole-like pockets centered at $\Gamma_0$ and $K_0$, and a dumbbell-shaped electron pocket around $M_0$. These Fermi surfaces originate

from the bands labeled $\alpha$ and $\beta$, respectively, which are a pair of spin-split bands due to broken inversion symmetry[22]. The charge-density-wave (CDW) order with a $3 \times 3$ superlattice modulation below $T_{CDW} \approx 75$ K[23,24] opens gaps on the $\beta$ Fermi surface (marked by the arrow), in agreement with density functional theory[2,25–27]. In the case of $Co_{0.32}TaS_2$, two Fermi surfaces are observed (see Fig. 1f and Supplementary Fig. S1), and are associated with different surface terminations (TaS$_2$- and Co-terminations). We differentiate these two terminations with spatially resolved core-level spectroscopy and the respective band structures along the high-symmetry directions as shown in Supplementary Fig. S1. This is similar to the V atoms doped NbS$_2$, where the V-terminations leads to more electron-doped band structures than the TaS$_2$-termination[28]. All band structures presented here are from the TaS$_2$-terminated surface, as it provides much sharper band structures. We next describe the evolution of the electronic structure upon Co intercalation. In detail, the $\alpha$ hole-like pocket centered at $\Gamma_0$ is significantly reduced in size compared to that of the undoped sample. In contrast, the $\alpha$ pocket at $K_0$ exhibits a much smaller shrinkage from 2$H$-TaS$_2$ to $Co_{0.32}TaS_2$. This suggests that the intercalation causes pocket-selective doping of the Fermi surfaces. In addition, the Fermi surface associated with the $\beta$ band evolves from a dumbbell-shaped electron pocket around $M_0$ in 2$H$-TaS$_2$ to two rounded hole-like pockets centered at $\Gamma_0$ and $K_0$. Most interestingly, upon Co intercalation, a shallow trigonally-warped electron-like pocket $\gamma$ appears at the corners of a $\frac{1}{\sqrt{3}} \times \frac{1}{\sqrt{3}}$ hexagonal Brillouin zone (dashed lines in Fig. 1f), whose orientation is rotated by 30° relative to that of the original $1 \times 1$ Brillouin zone of 2$H$-TaS$_2$. This pocket likely originates from Co-derived bands, as it is centered at the corners of the Co-lattice Brillouin zone. For simplicity, we refer to these bands as the Co-derived bands hereafter. However, the $\gamma$ pocket is absent in the Co-terminated surface (see Supplementary Fig. S1). One might wonder why the Co-terminated surface did not show the Co-derived band. A scanning tunneling microscopy (STM) study on a similar material, $Cr_{1/3}NbS_2$, provided a plausible explanation: an ideal cleavage is expected to split the Co intercalants equally between the two cleaved surfaces, resulting in a disordered Co configuration, which renders the Co-derived bands ill-defined[29].

Figure 1g, h compare the band structures of undoped and Co-intercalated samples along the high-symmetry direction $M_0 - \Gamma_0 - K_0$. The direct comparison reveals two striking differences: (i) The $\alpha$ and $\beta$ bands are significantly electron-doped in $Co_{0.32}TaS_2$, signifying that Co acts as an electron dopant by transferring some of its 3$d$ electrons to the TaS$_2$ layers. (ii) A shallow electron-like band emerges at K ($\gamma_K$) together with a near-$E_F$ hole-like band at M ($\gamma_M$), giving rise to the triangular Fermi pockets ($\gamma$).

These findings are quantified in Fig. 1j, k, where the experimental band dispersions of 2$H$-TaS$_2$ and $Co_{0.32}TaS_2$ were obtained by fitting the positions of the Lorentzian-shaped peaks of the momentum distribution curves (MDCs) and energy distribution curves (EDCs), as shown in Supplementary Fig. S2. The fitting results reveal the following effects of Co intercalation: (i) Co not only electron-dopes the spin-orbit-split $\alpha$ and $\beta$ bands overall, but also modifies their dispersions anisotropically, indicating non-rigid shifts. Quantitatively, along the $\Gamma_0 - K_0$ direction, the shift of the $\alpha$ band minimum is ≈260 meV, while along the $\Gamma_0 - M_0$ direction the shifts of the $\alpha$ and $\beta$ bands are much larger (≈340 meV at the band bottom and ≈360 meV at $M_0$), as extracted from EDC spectra in Supplementary Fig. S3. Density Functional Theory (DFT) calculations for 2$H$-TaS$_2$ reveal that the observed ~300 meV band shift can be reproduced by adding one electron per TaS$_2$ unit cell, in good agreement with a simple electron counting picture of $Co_{1/3}TaS_2$ consisting of alternating $[Co_{1/3}]^+$ and $[TaS_2]^-$ layers (see Supplementary Fig. S4). (ii) Spin-orbit coupling is enhanced, as supported by the enlarged momentum splitting between the $\alpha$ and $\beta$ bands from 0.06 to 0.18 Å (see MDCs at $E_F$ in Fig. 1j, k, and Supplementary Fig. S3)[30]. (iii) A shallow electron-like band ($\gamma_K$) appears at K

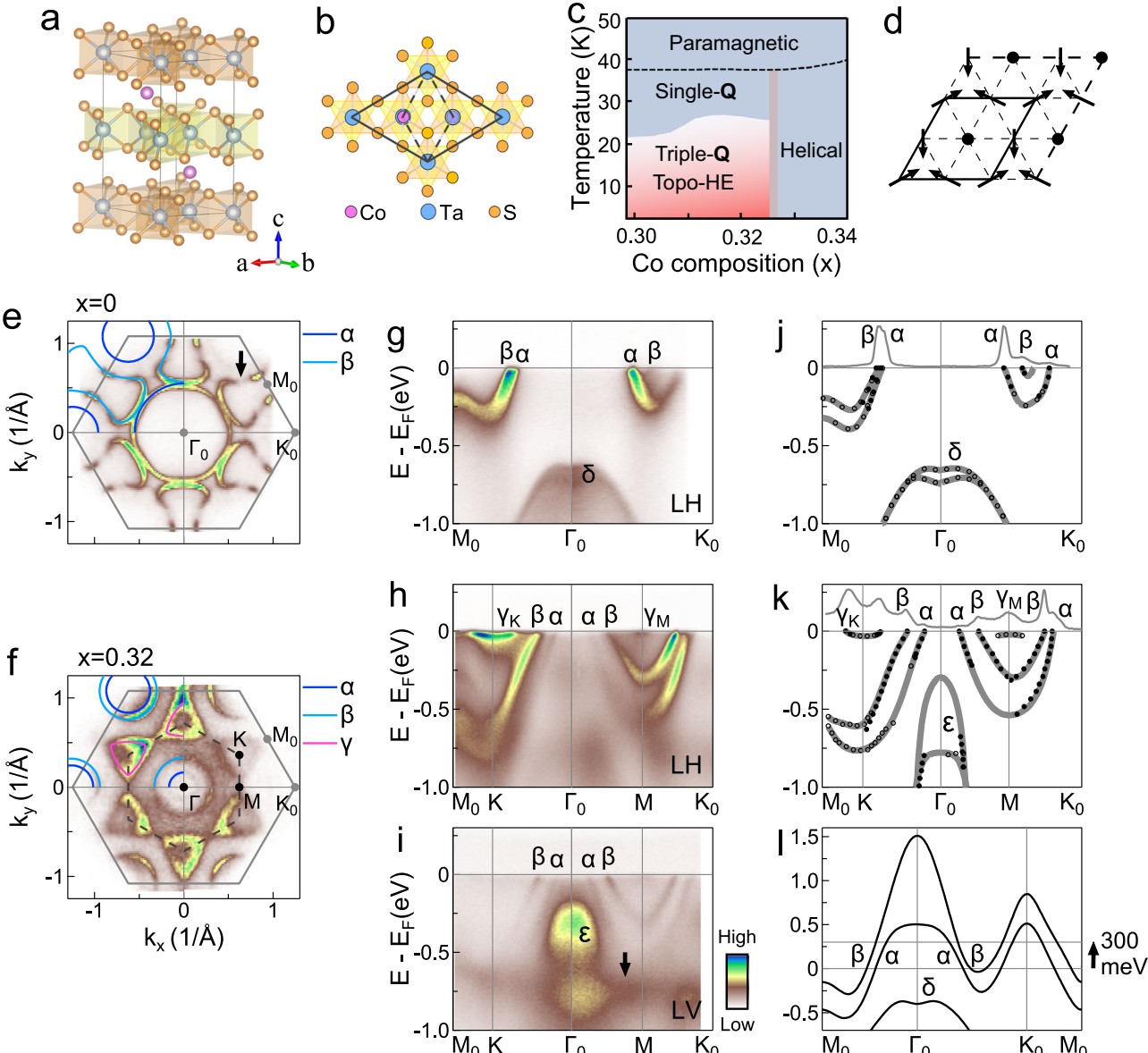

**Fig. 1 | Electronic structure comparison between undoped and Co-doped 2H-TaS₂. a** Crystal structure of Co$_{1/3}$TaS₂. Adjacent TaS₂ layers with opposite in-plane orientations are highlighted in orange and yellow. **b** Top view of the crystal structure. The solid and dashed black lines represent the unit cells of Co$_{1/3}$TaS₂ and 2H-TaS₂, respectively. **c** Phase diagram of Co$_{1/3}$TaS₂, adapted from ref. 20. **d** Top view of the 3**Q** magnetic order on a triangular lattice. The black lines indicate the magnetic unit cell. **e, f** Low temperature (7 K) Fermi surfaces mapping of 2H-TaS₂ and Co$_{0.32}$TaS₂. The solid (dashed) lines denote the Brillouin zone of 2H-TaS₂ (Co$_{1/3}$TaS₂). High-symmetry points are labeled as $\Gamma_0$, $K_0$, and $M_0$ for 2H-TaS2, and $\Gamma$, K, and M for Co$_{1/3}$TaS₂. The gapped portion indicated by an arrow signifies the charge density wave (CDW) gap in 2H-TaS₂. In both panels, the fitted Fermi surfaces obtained from the raw data are overlaid in the upper-left quadrants. Observed Fermi surface sheets are marked with $\alpha$ (dark blue curves), $\beta$ (light blue curves) and $\gamma$ (magenta curves). Energy-momentum intensity plots of 2H-TaS₂ (**g**) and Co$_{0.32}$TaS₂ (**h, i**), measured along the $M_0$-$\Gamma_0$-$K_0$ high-symmetry direction. Data in (**g, h**) are acquired using linearly horizontally (LH) polarized light, whereas (**i**) uses linearly vertically (LV) polarized light. **j, k** Extracted band dispersions of 2H-TaS₂ ($\alpha$, $\beta$ and $\delta$) and Co$_{0.32}$TaS₂ ($\alpha$, $\beta$, $\gamma_K$, $\gamma_M$ and $\epsilon$), together with the corresponding momentum distribution curves (MDCs) at the Fermi level ($E_F$). Filled and open circles denote the peak positions of MDCs and energy distribution curves (EDCs), respectively, extracted from Supplementary Fig. S2. **l** Calculated band structures including spin-orbit coupling along $M_0$-$\Gamma_0$-$K_0$ in the $k_z = 0$ plane for bulk 2H-TaS₂. The Fermi level of Co$_{0.32}$TaS₂ is referenced by shifting $E_F$ upward by 300 meV.

within ~40 meV of $E_F$ (Fig. 1h, k). (iv) A hole-like feature ($\gamma_M$) is observed around M within ~30 meV of $E_F$ (see also Fig. 1h, k). This result is reminiscent of the magnetically intercalated NbS₂ compound, where a shallow electron pocket develops due to the Co interstitials[31,32]. While it may be tempting to attribute the $\gamma_K$ and $\gamma_M$ bands to unoccupied states of pristine TaS₂, band-structure calculations for 2H-TaS₂ show no evidence of such states (see Fig. 1l). We further performed DFT calculations for Co$_{1/3}$TaS₂ and identified highly-dispersive bands primarily derived from Ta orbitals, together with near-$E_F$ flat bands mainly derived from Co orbitals (see Supplementary Fig. S5). This result

strongly supports the conclusion that the $\gamma_K$ and $\gamma_M$ bands arise from the Co intercalants. (v) In addition to these near $E_F$ bands, a highly dispersive band around $\Gamma_0$ ($\epsilon$) emerges.

To gain further insights into the orbital characters of these bands, Fig. 1i presents the band dispersions of Co$_{0.32}$TaS₂ measured using linearly vertically (LV) polarized light. Compared with the linearly horizontally (LH) data (Fig. 1h), the LV spectra show three main differences: the disappearance of the near-$E_F$ $\gamma_K$ and $\gamma_M$ bands, a pronounced reduction of spectral weight in the highly-dispersive $\alpha$ and $\beta$ bands, and a significant enhancement of the spectral intensity of the $\epsilon$

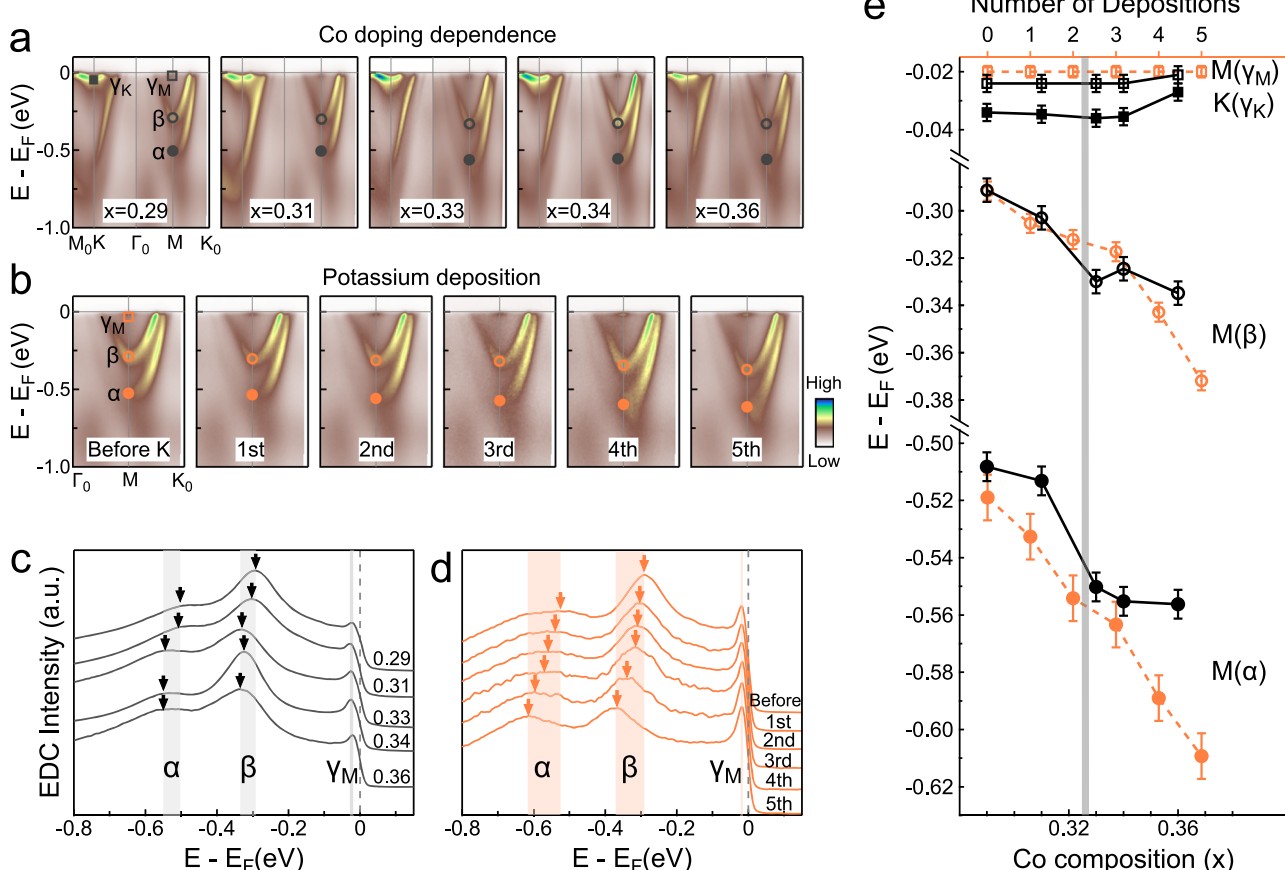

**Fig. 2 | Co doping and potassium (K)-deposition effects on the electronic structure of Co$_x$TaS$_2$. a** ARPES spectra along the M$_0$-Γ$_0$-K$_0$ high-symmetry direction for Co$_x$TaS$_2$ samples with $x$ = 0.29, 0.31, 0.33, 0.34, and 0.36. **b** ARPES spectra of Co$_{0.29}$TaS$_2$ along the Γ$_0$-K$_0$ direction before and after successive potassium (K) depositions. **c** EDCs at the M point for Co$_x$TaS$_2$ samples with $x$ = 0.29, 0.31, 0.33, 0.34 and 0.36, obtained from (**a**). **d** EDCs at the M point for Co$_{0.29}$TaS$_2$ before and after multiple rounds of K deposition, obtained from (**b**). EDC peaks corresponding to the $\alpha$, $\beta$, and $\gamma_M$ bands are labeled accordingly. **e** Evolution of the band bottoms of the $\alpha$, $\beta$, $\gamma_K$, and $\gamma_M$ bands as a function of Co doping and K deposition. Black and orange curves labeled M($\gamma_M$), M($\beta$), and M($\alpha$) represent the energy positions of the $\gamma_M$, $\beta$, and $\alpha$ peaks, respectively, obtained from (**c**, **d**). The curve labeled K ($\gamma_K$) shows the Co-doping dependence of the $\gamma_K$ band bottom, extracted from EDCs at the K point (see Supplementary Fig. S8). Error bars are determined based on the fitting error of the EDC peak position.

band. Based on the matrix-element analysis and the DFT-calculated orbital projected band structures (see Supplementary Note 1 and Supplementary Figs. S6, S7)[33,34], we infer that the near-E$_F$ $\gamma_K$ and $\gamma_M$ bands mainly originate from Co 3$d_{z^2}$ orbitals, the highly-dispersive $\alpha$ and $\beta$ bands are dominated by Ta 5$d_{z^2}$ and Ta 5$d_{xy}/d_{x^2-y^2}$ orbitals, and the $\epsilon$ band is primarily contributed by $d_{xz}/d_{yz}$ and $d_{xy}/d_{x^2-y^2}$ states. In addition, the LV spectra reveal a non-dispersive band near −0.75 eV (see the arrow in Fig. 1i), which could be associated with an impurity-like localized state.

## Evolution of band structure with Co doping and potassium deposition

Figure 2 summarizes the electron-doping effect produced by varying Co composition and by successive potassium depositions. Figure 2a presents the Co-doping-dependent spectra along the high symmetry direction M$_0$ − Γ$_0$ − K$_0$, revealing a continuous downward-shift of the $\alpha$ and $\beta$ bands with increasing Co doping. Consistent with this observation (Fig. 2c), the peak positions of the raw EDCs at M, which mark the bottoms of the $\alpha$ and $\beta$ bands, shift toward higher binding energy as the doping increases from $x$ = 0.29 to 0.36. The overall energy shift of the $\alpha$ and $\beta$ bands is ~50 meV. The observed doping dependence of the $\alpha$-band minimum is well reproduced by a rigid-band-shift approximation applied to the TaS$_2$ band structure (see Supplementary Fig. S4). In contrast, the extrema of

the $\gamma_M$ and $\gamma_K$ bands exhibit only a weak doping dependence (Fig. 2a). In line with this, the near-E$_F$ EDC peaks of the $\gamma_M$ and $\gamma_K$ bands move only slightly (Fig. 2c and Supplementary Fig. S8). Specifically, as indicated by the black curves in Fig. 2e, small upturns (~3 and ~8 meV for the $\gamma_M$ and $\gamma_K$ bands, respectively) are observed at $x$ = 0.36. Taken together, these results indicate that Co doping induces a much larger electron-doping effect on the TaS$_2$-derived bands than on the Co-derived bands.

To validate the intrinsic nature of the observed doping dependence and eliminate extrinsic sources, such as inhomogeneous doping level or variations in sample quality across different doping levels, we report in Fig. 2b the evolution of the band structure upon tuning the chemical potential on the same sample. This is achieved via in situ doping by depositing of alkali-metal atoms, a technique widely used in ARPES to study doping-dependent electronic structures in a variety of materials[35]. The alkali metal atoms donate electrons due to their low ionization potential, causing the exposed surface to be electron-doped. The EDCs obtained at M for Co$_{0.29}$TaS$_2$ before and after multiple rounds of potassium (K) depositions are shown in Fig. 2d, and the band minima as a function of K dosing are represented by the orange curves in Fig. 2e. Specifically, after four rounds of K deposition, we observe a downshift of the $\alpha$ and $\beta$ bands comparable to that induced by increasing the Co doping from $x$ = 0.29 to 0.36. Moreover, the $\gamma_M$ band remains nearly unchanged within the experimental resolution.

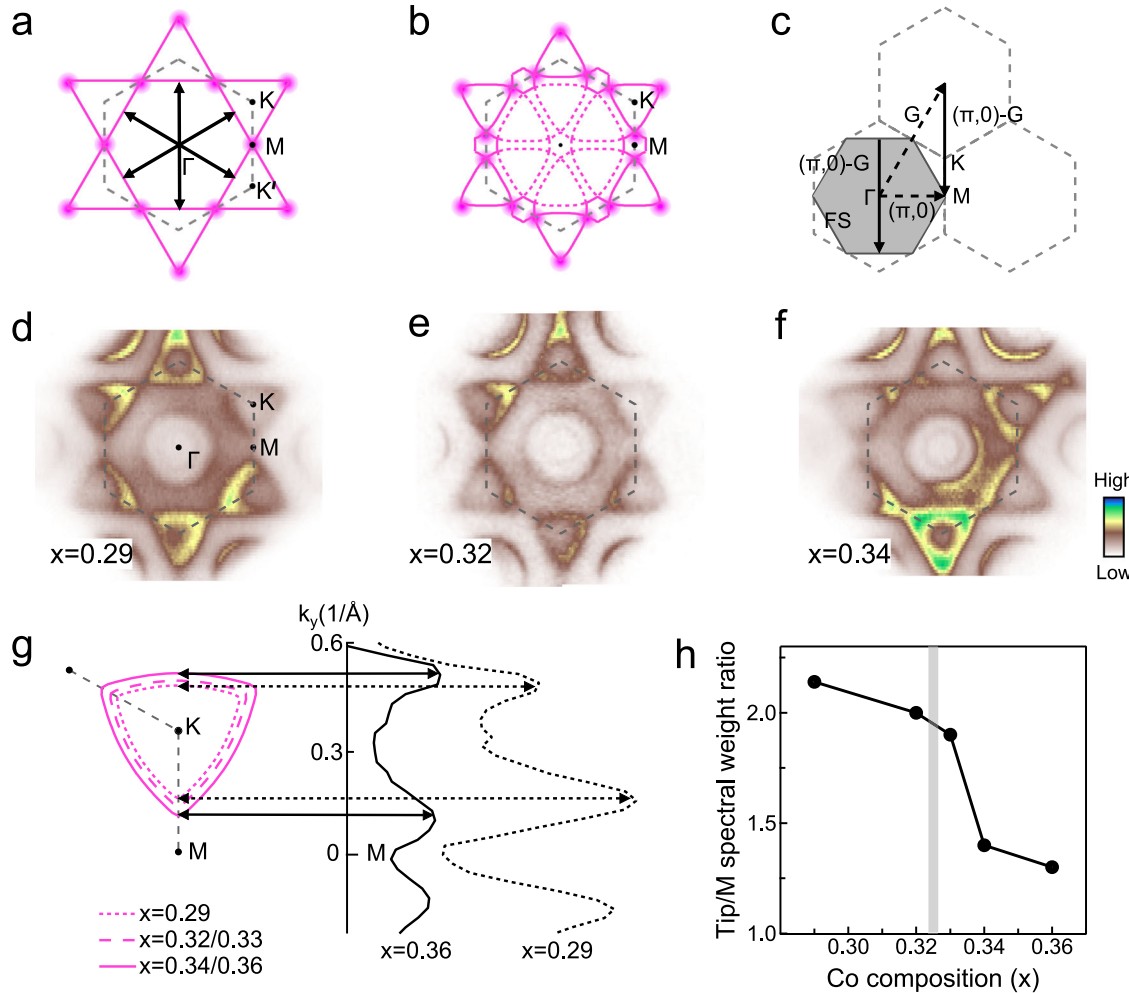

**Fig. 3 | Doping-dependent Fermi surfaces of Co$_x$TaS$_2$ and signature of a phase transition across $x_c$. a** Calculated Fermi surface of a triangular lattice at 3/4-filling, without the 3**Q** magnetic order. The edge centers of the hexagonal Fermi surface are connected by three nesting wave vectors. Van Hove singularities (VHSs) located at the M points are marked by magenta dots. **b** Calculated Fermi surface of a triangular lattice slightly away from 3/4-filling, incorporating 3**Q** magnetic order. In this case, the VHSs are located at the tips of the triangular Fermi pockets. Original and folded Fermi surface sheets are shown as solid and dashed magenta curves, respectively. Computational details are provided in the Methods section. **c** The nesting wave vector (($\pi$,0)-**G**) connects the edges of the hexagonal Fermi surface at 3/4-filling, and is equivalent to ($\pi$, 0) up to a reciprocal lattice vector **G**. The corresponding 3**Q** magnetic order in real space and the reconstructed Brillouin zone are illustrated in Supplementary Fig. S9. **d–f** Experimental Fermi surface maps for samples with $x = 0.29$, $x = 0.32$, and $x = 0.34$, respectively. **g** Evolution of the triangular Fermi pockets as a function of doping level $x$, extracted from the raw spectra (Supplementary Fig. S10). MDCs obtained along the K-M direction for $x = 0.29$ and $x = 0.36$ samples are shown on the right. **h** Ratio of spectral weight on the Fermi surface between the tip of the $\gamma$ pocket and the M point as a function of Co composition $x$.

These findings lead to two key insights: (i) The fact that the Co-derived $\gamma_M$ band barely shifts upon K dosing suggests a high density of states. (ii) Increasing the Co doping to $x = 0.36$ causes a small upward shift in the $\gamma_K$ and $\gamma_M$ bands, in contrast to the nearly unchanged band extrema observed under K deposition. This suggests that Co doping affects the Co-derived $\gamma_K$ and $\gamma_M$ bands beyond simple electron doping. This motivates the question: could this be a manifestation of changes in magnetic order as a function of Co doping?

**Signature of 3Q magnetic order on the Fermi surface**

A tempting assertion is that the effects on the $\gamma_K$ and $\gamma_M$ bands stem from changes in the non-coplanar (3**Q**) magnetic order. Since DFT struggles to accurately describe the dispersion of the near-E$_F$ flat bands, primarily due to strong electron-electron correlations and the presence of local magnetic moments, it cannot provide a reliable description of our system. At the same time, the near-E$_F$ flat bands are dominated by Co 3$d$ orbitals, whereas the dispersive $\alpha$ and $\beta$ bands mainly originate from TaS$_2$-derived states (see Supplementary Fig. S5).

Given both the limitations of DFT and the distinct orbital characters of these states, we adopt a phenomenological tight-binding model on a Co triangular lattice to account for the flat bands and their coupling to local magnetic moments. We simulate the Co-derived bands using a $\sqrt{3} \times \sqrt{3}$ triangular lattice model formed by the Co atoms, as illustrated in Fig. 1. In the absence of the 3**Q** magnetic order—that is, in the nonmagnetic state (exchange coupling constant $J = 0$)—the Fermi surface of the 3/4-filled band with nearest-neighbor hopping is shown in Fig. 3a. Here, the triangular electron-like pockets centered at K touch each other at M, where a van Hove singularity (VHS) with a divergent density of states is located. The existence of a VHS can naturally explain the nearly electron-doping-independent nature of the $\gamma_K$ and $\gamma_M$ bands, observed in Fig. 2. At 3/4-filling, the 3**Q** magnetic order is predicted to emerge due to Fermi surface nesting[36,37]. The nesting wave vectors are shown by the black solid arrows in Fig. 3a, c (see Supplementary Fig. S9). In the presence of the 3**Q** magnetic order, two significant modifications to the Fermi surface occur (Fig. 3b): (i) When the Fermi level passes through the band top, the triangular Fermi

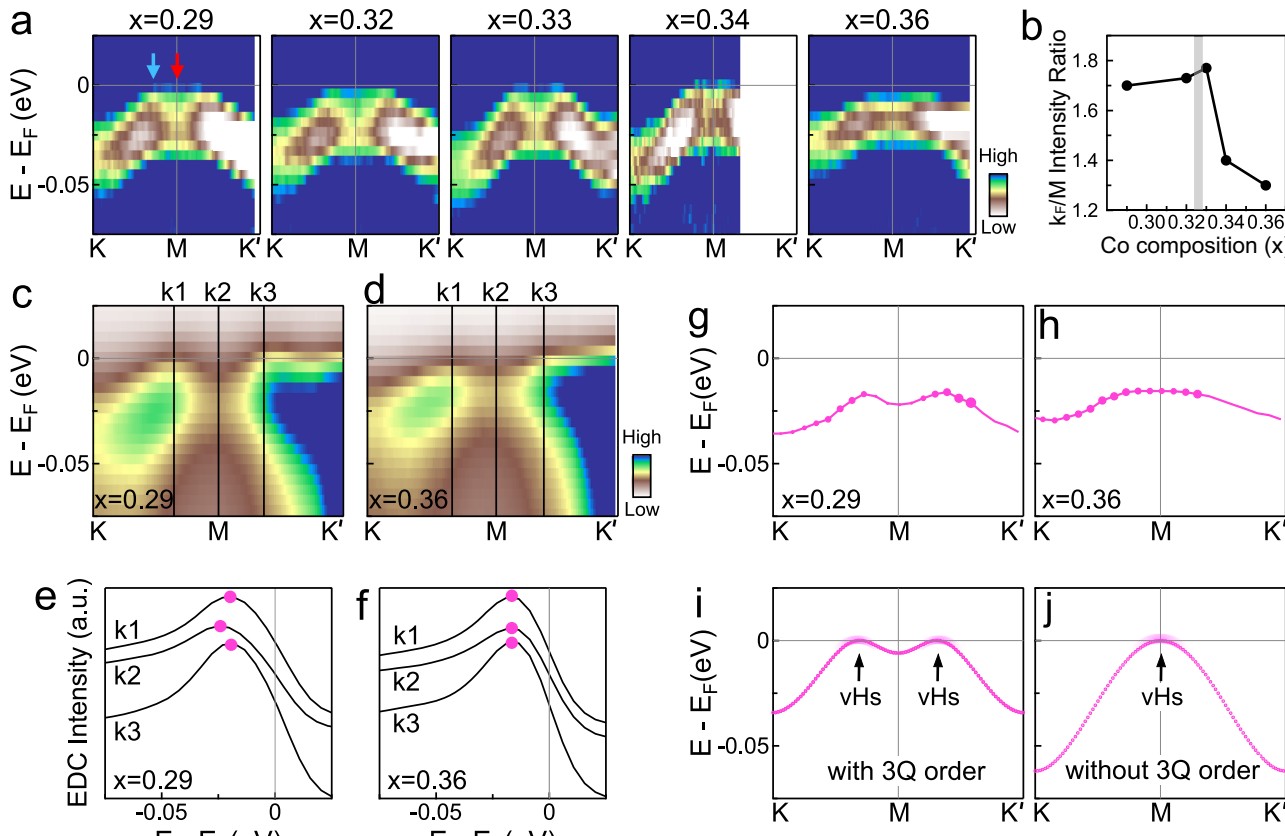

**Fig. 4 | Doping-dependent band structure along K-M-K′ and signature of the 3Q states for $x < x_c$. a** Second-derivative images (with respect to energy) along the K-M-K′ direction as a function of Co doping, derived from the raw data shown in Supplementary Fig. S11. The blue and red arrows indicate the momenta corresponding to the Fermi momentum ($k_F$) of the $\gamma_K$ band and the M point in the $x = 0.29$ sample. **b** Intensity ratio of EDC peaks obtained at $k_F$ and M in (**a**), plotted as a function of Co composition $x$. Experimental band structures for the $x = 0.29$ (**c**) and $x = 0.36$ (**d**) samples along the K-M-K′ direction. **e, f** Representative EDCs extracted at three selected momenta (k1, k2, and k3), as marked by black lines in (**c, d**). **g, h** Experimentally extracted band dispersion along the K-M-K′ direction for the

$x = 0.29$ (**g**) and $x = 0.36$ (**h**) samples, obtained from EDC stacks provided in Supplementary Fig. S11. The dot size reflects the relative intensity of the EDC peaks, highlighting the distribution of spectral weight. **i, j** Calculated band structures along the K-M-K′ direction for a triangular lattice with (**i**) and without (**j**) 3Q order. In the presence of 3Q order, two VHSs appear on either side of the M point, accompanied by a band dip at M. In contrast, without 3Q order, a single VHS is located at M. Calculated band structures along the Γ-M-Γ direction are provided in Supplementary Fig. S13. Details of the computational parameters are provided in the Methods section.

surface shows a notable shrinkage, as a single VHS at M evolves to two VHSs located away from M. (ii) The magnetic order enlarges the unit cell and shrinks the Brillouin zone, inducing band folding from M to Γ (see Supplementary Fig. S9).

Figure 3d–f shows the experimental Fermi surfaces for three doping levels: $x = 0.29$, 0.32 (both below $x_c$), and $x = 0.34$ (above $x_c$). Additional Fermi surfaces for other doping levels are provided in Supplementary Fig. S10. In samples with $x < x_c$, i.e., in the presence of the 3Q magnetic order, no signatures of band folding are observed. This absence could be attributed to the coherence factor, which dramatically weakens the intensity of the folded bands—a phenomenon commonly seen in many reconstructed Fermi surfaces[38]. The comparison shows a clear enlargement of the triangular Fermi pockets ($\gamma$) as the doping increases. This observation is illustrated in Fig. 3g, where the $\gamma$ pockets are extracted from the experimental Fermi surfaces for all doping levels. Such enlargement is evident in the MDCs at $E_F$ along the K-M direction, where the peak-to-peak distance clearly increases as the doping rises from $x = 0.29$ to $x = 0.36$. This trend is further evidenced by the variation in the relative spectral weight on the Fermi surface between the tip of the $\gamma$ pocket and the M point, as shown in Fig. 3h, where a transition across $x_c$ is observed. We attribute the observed enlargement of the $\gamma$ pockets to the weakening of the 3Q magnetic order above $x_c$, rather than to increased Co doping. This

assignment is supported by the observation that the $\gamma_K$ band bottom does not shift downward with doping (see Fig. 2), which is inconsistent with a simple band-filling picture. Instead, the weakening of the 3Q order alters the Fermi surface topology, naturally leading to the enlarged $\gamma$ Fermi pockets.

### Signature of 3Q magnetic order on the band dispersions

Further evidence of the 3**Q** magnetic order is provided by the near-$E_F$ band dispersion along the K-M-K′ direction. The second derivative images of the experimental spectra for different doping levels across $x_c$ are shown in Fig. 4a. Two clear qualitative changes are observed near the M point as the doping increases: (i) an increase in the spectral weight at the M point relative to the Fermi momentum $k_F$, as indicated by the red and blue arrows in the leftmost panel; (ii) a change of curvature around M from electron-like to hole-like. To quantify the first feature, we extract the momentum-resolved EDCs, as shown in Supplementary Fig. S11. In Fig. 4b, we plot the doping-dependent ratio of EDC peak intensities at $k_F$ and at M. An obvious decrease in the ratio across $x_c$ suggests a spectral weight transfer from the tip of the $\gamma$ pocket to the M point. To quantify the second feature, in Fig. 4c, d and e, f we show zoom-in views of the raw spectra and representative EDCs for $x = 0.29$ and 0.36, respectively. For $x = 0.29$, a shallow but distinct dip in the dispersion is observed at M, accompanied by two

band tops nearby. This is even more evident in the EDCs where the peak position at the M point appears at a lower energy than in the EDCs away from M (Fig. 4e), in contrast to the spectra for $x = 0.36$, where the EDC at M has the same peak position as other EDCs (Fig. 4f). The extracted dispersions in Fig. 4g, h clearly summarize these observations, showing an evolution from an inverse Mexican-hat-like dispersion in $x = 0.29$ to a hole-like dispersion in $x = 0.36$. To understand the origin of such evolution, in Fig. 4i, j, we report the calculated band dispersion along K-M-K′ with and without the 3**Q** order, respectively. The calculation reveals that, in the absence of the 3**Q** order, the dispersion is hole-like, and hence there is a single VHS at the M point (Fig. 4j). In contrast, with the 3**Q** order, the dispersion evolves into an inverse Mexican-hat-like shape, characterized by two VHSs positioned away from M along the K-M-K′ direction (Fig. 4i). We further tested the robustness of this result by varying hopping amplitude $t$ and coupling constant $J$, and found that the inverse Mexican-hat-like dispersion persists as long as the 3**Q** order is present (see Supplementary Fig. S12). Our experimental band dispersion along K-M-K′ for samples with $x < x_c$ closely resembles the inverse Mexican-hat-like dispersion predicted in Fig. 4i, strongly suggesting that the band modification is induced by the 3**Q** magnetic order.

It is noteworthy that the band dispersions for samples with $x > x_c$ (Fig. 4h) cannot be simply interpreted using the simulated band structure without the 3**Q** order (Fig. 4j), as a reliable prediction of the electronic structure in this region is hindered by the uncertainty in the moment arrangement of the "helical" magnetic order. Nonetheless, several key features serve as circumstantial evidence for the alleged transition from 3**Q** to helical magnetic order. First, the dispersion dip at M is completely suppressed in $x = 0.34$ and 0.36 (see Supplementary Fig. S11 for raw spectra and EDC stacks at all dopings). Second, the spectral-weight transfer, as summarized in Figs. 3h and 4b, from the tip of the triangular pocket for $x < x_c$, to the M point for $x > x_c$, is suggestive of the displacement of the VHSs from Fig. 4i to j. Although the electronic structure in samples with $x > x_c$ can not be adequately described by the model without incorporating 3**Q** order, our results, when taken at face value, qualitatively align with the doping-tuning experiments conducted via Co composition variation[20] and ionic gating[39]. Namely, $x = 0.29$ and $x = 0.33$ mark the onset of two distinct magnetic ground states: a tetrahedral 3**Q** magnetic structure accompanied by a pronounced topological Hall effect, and a helical magnetic order that lacks the topological Hall effect, respectively.

Temperature-dependent measurements provide further evidence for the role of 3**Q** magnetic order. As shown in Supplementary Fig. S14 and discussed in Supplementary Notes 2 and 3, the inverse Mexican-hat-like dispersion is observed only in the 3**Q** ordered phase at low temperature. At higher temperatures, corresponding to the 1**Q** ordered and paramagnetic states, the dispersion evolves into a simple hole-like band along the K-M-K′ direction. These observations are in close agreement with tight-binding calculations, which show that only the 3**Q** ordered state gives rise to the inverse Mexican-hat-like dispersion (Fig. 4 and Supplementary Fig. S15). Together, these results demonstrate that the reconstruction of the near-$E_F$ bands is a distinctive fingerprint of the 3**Q** order.

In summary, we investigated the electronic structures of 2$H$-TaS$_2$ and Co$_x$TaS$_2$ as a function of Co doping within the range of $x = 0.29$-0.36. A comparison between 2$H$-TaS$_2$ and Co$_{0.32}$TaS$_2$ shows that Co intercalation injects electrons into the parent TaS$_2$ layers and leads to the emergence of Co-derived bands near the Fermi level. In addition, our experiment provides vivid evidence of the 3**Q** magnetic order, namely an inverse Mexican-hat-like dispersion around M, and the associated high density of states regions in momentum space. Moreover, the spectral-weight transfer is suggestive of a transition from 3**Q** to helical magnetic order. Key open questions include the precise arrangement of magnetic moments in the helical magnetic order and the corresponding predictions for the electronic structure,

as well as the discrepancy between experiment and theory in the dispersion of the $\gamma_M$ band (see Supplementary Fig. S16). Nevertheless, the present results illustrate the effect of an underlying 3**Q** magnetic order, motivating the exploration of this class of systems as potential hosts for a quantized quantum anomalous Hall effect at 3/4-filling.

## Methods

### Growth and characterization of single crystals
High-quality single crystals of Co$_x$TaS$_2$ were grown by a two-step procedure. First, a precursor was prepared. The elements were combined in a ratio of Co:Ta:S ($x$:1:2), where $x = 0.29$-0.36 in different growth batches. These batches were separately loaded in alumina crucibles and sealed in quartz tubes under a partial pressure (200 torr) of Argon gas. The tube was heated to 900 °C and kept there for 10 days. The furnace was then shut off and allowed to cool naturally. This reaction yields a free-flowing shiny powder of polycrystals that was ground separately for each batch. Second, the precursor was loaded with iodine in a quartz tube, evacuated, and placed in a horizontal two-zone furnace. The precursor and iodine were placed in zone 1 and the other end of the tube (the growth zone) was in zone 2. Both zones were heated to 850 °C for 6 h to encourage nucleation. Then, zone 1 was raised to 950 °C while zone 2 was kept at 850 °C. This condition was maintained for 10 days. The furnace was then shut off and allowed to cool naturally. Shiny hexagonal crystals up to 1 cm in lateral length were collected from the cold zone. Energy dispersive spectroscopy (Xplore 30, Oxford Instruments) was used to determine the Co composition for each sample. For each single crystal, 10 different regions were measured, and their average value and standard deviation were used. 2$H$-TaS$_2$ single crystals were purchased from 2D Semiconductors.

### High-resolution ARPES measurements
Angle-resolved photoemission measurements were performed at Beamline 7.0.2 (MAESTRO) of the Advanced Light Source, equipped with an R4000 hemispherical electron analyzer (Scienta Omicron). Data for the Co-intercalated samples were taken with $hv = 79$ eV under two different light polarizations: linear horizontal (LH) and linear vertical (LV). For the pristine 2$H$-TaS$_2$ compound, measurements were conducted at $hv = 93$ eV and with LH-polarized light. All experiments were carried out at low temperature ($T = 7$ K). Preliminary data were also collected at Beamline 10.0.1.2 of the Advanced Light Source.

### Theory
The Co-derived bands are modeled using the following Hamiltonian defined on the Co triangular superlattice.

$$H = -t \sum_{\langle ij \rangle, \alpha} c_{i\alpha}^\dagger c_{j\alpha} - \mu \sum_{i, \alpha} c_{i\alpha}^\dagger c_{i\alpha} - J \sum_i \mathbf{S}_i \cdot c_{i\alpha}^\dagger \boldsymbol{\sigma}_{\alpha\beta} c_{i\beta} \tag{1}$$

Here, $t$ represents the nearest-neighbor $\langle ij \rangle$ hopping matrix element. $J$ denotes the coupling constant between the itinerant electrons (annihilated by $c_{i\alpha}$) and the classical spins $\mathbf{S}_i$ associated with the 3**Q** ordering. $\mu$ is the chemical potential. In the absence of the 3**Q** order, we set $t = -0.062$, $\mu = 2t$, and $J = 0$. In the presence of the 3**Q** order, the parameters are $t = -0.062$, $\mu = 2.545t$, and $J = 0.04$.

To examine the 1**Q** ordered state, we further constructed a tight-binding Hamiltonian on the $2 \times 1$ supercell.

$$H = -t_1 \sum_{\langle ij \rangle_1, \alpha} c_{i\alpha}^\dagger c_{j\alpha} - t_2 \sum_{\langle ij \rangle_2, \alpha} c_{i\alpha}^\dagger c_{j\alpha} - t_3 \sum_{\langle ij \rangle_3, \alpha} c_{i\alpha}^\dagger c_{j\alpha}$$
$$- \mu \sum_{i, \alpha} c_{i\alpha}^\dagger c_{i\alpha} - J \sum_i \mathbf{S}_i \cdot c_{i\alpha}^\dagger \boldsymbol{\sigma}_{\alpha\beta} c_{i\beta} \tag{2}$$

The parameters are chosen as $t_1 = 0.08$, $t_2 = -0.04$, $t_3 = -0.005$, and $\mu = -2.85\,t_1$. Calculations show that the band dispersions remain hole-like along the K-M-K′ direction for all tested $J$, indicating that the

presence or absence of the 1Q magnetic order does not qualitatively alter the dispersion (see Supplementary Fig. S15).

## DFT calculations

We performed first-principles calculations using the Vienna ab initio simulation package (VASP)[40] within the Perdew-Burke-Ernzerhof (PBE) generalized gradient approximation[41] for exchange-correlation and projector-augmented-wave (PAW) potentials. A plane-wave basis set with a cutoff energy of 400 eV was utilized. Spin-orbit coupling was included via the second-variation method throughout. For $Co_{0.33}TaS_2$, we employ a $2 \times 2 \times 1$ supercell that hosts the 3Q non-coplanar anti-ferromagnetic order[19], with initial magnetic moment of $2.25 \mu_B$ for each Co atom. Electronic self-consistency used a $\Gamma$-centered $7 \times 7 \times 7$ k-mesh for the supercell, an electronic self-consistency energy convergence of $10^{-6}$ eV, and Gaussian smearing with a broadening parameter of 0.04 eV. To capture the strong electron correlation effects of localized Co $3d$ orbitals, we used the DFT + U approach[42] with an effective Hubbard $U = 4.1$ eV and $J_{Hund} = 0.8$ eV, as determined by the constrained RPA method[43]. We obtained and used the optimized crystal structure with the force criterion for the relaxation fixed at 1 meV/Å with 3Q antiferromagnetic order. We performed post-processing, including band plotting, band unfolding and orbital-resolved projections using VASPKIT tools[44]. To compute the density of states for $2H$-$TaS_2$, we used a dense $k$-grid of $21 \times 21 \times 10$ in the primitive Brillouin zone and the tetrahedron method.

## Data availability

The ARPES data generated during this study have been deposited in the Zenodo database under accession code: https://doi.org/10.5281/zenodo.18285547.

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

## Acknowledgements

We thank Sang-Wook Cheong for insightful discussions. We thank Cheng Hu for taking the data for 2*H*-TaS$_2$. This work was primarily supported by the U.S. Department of Energy, Office of Science, Office of Basic Energy Sciences, Materials Sciences and Engineering Division, under contract No. DE-AC02-05-CH11231 (Quantum Materials Program KC2202). This research used resources of the Advanced Light Source, a US DOE Office of Science User Facility under Contract No. DE-AC02-05CH11231. G.C. and D.D. acknowledge support from the National Research Foundation, Singapore, under its Fellowship Award (NRFNRFF13-2021-0010) and from the Singapore Ministry of Education (MOE) Academic Research Fund Tier 3 grant (MOEMOET32023-0003).

## Author contributions

A.L. and H.-L.L. conceived this project. H.-L.L. performed the ARPES experiments with assistance from M.H., H.J., L.M., A.F., C.J., and A.B. and analyzed the resulting data. J.R., C.X., and J.A. contributed to crystal synthesis. D.D. and G.C. contributed to DFT calculations. D.-H.L. contributed to theoretical work. H.-L.L., D.-H.L., and A.L. wrote this paper. All authors participated in discussions and provided comments on the paper.

## Competing interests

The authors declare no competing interests.
