## [Peer Review File · Nature Communications]

Discovery of Van Hove Singularities: Electronic Fingerprints of 3Q Magnetic Order in a Van der Waals Quantum Magnet

Corresponding Author: Professor Alessandra Lanzara

Version 0:

Reviewer comments:

Reviewer #1

(Remarks to the Author)

This manuscript reports the angle resolved photoemission spectroscopy (ARPES) study of Co_xTaS_2 providing the evidence of the electronic structure to support the sharp magnetic transition from a non-coplanar 3Q order to a helical order upon electron doping at low temperature. Co_xTaS_2 forms an intercalated Co triangular lattice between TaS₂ layers and the doping of Co ions can change the electronic structure dramatically near $x=0.33$, possibly accompanied by the magnetic transition. The main conclusion of this manuscript is that the electron doping effect mainly changes Ta band minimum as evidenced from the ARPES measurements of both Co dopings and K depositions, while the changes of Co-derived bands are evident from only the Co doping effect which is attributed to the spin-exchange interaction change of itinerant bands with localized spins. The manuscript is clearly written and address an important and interesting topic in these Van-der-Waals quantum magnets. While these findings can be of great interests to other similar intercalated transition metal dichalcogenides, I am not convinced that a rather simple tight-binding model that they used can explain the essential physics of Co-derived bands near the Fermi energy and the parameters they used are not justified.

1. To understand the ARPES band changes upon electron dopings, the authors used a rather simple tight-binding model including the nearest-neighbor hopping parameter t and the Kondo coupling J between itinerant electrons and the classical spins. However, there is no justification of the parameters they used. In my opinion, it would be beneficial to use some ab-initio calculations to verify the quantitative features of band structure changes near the Fermi energy. For example, the authors should at least check if the itinerant Co d band ($J=0$) is consistent with the Co projected DFT band structure of $\text{Co}_{1/3}\text{TaS}_2$. Moreover, these itinerant Co d_{z^2} bands are covalently bonded with Ta d_{z^2} bands. They should justify why these Ta d_{z^2} orbitals can be neglected in this simple tight-binding model. They should also check how sensitively the band structure near the M point changes due to the change of t and J parameters.

2. The authors argue that Fig.3a Fermi surface is obtained from the absence of the 3Q magnetic order- "namely, when the magnetic moments are randomly oriented". However, it looks like the Fermi surface can be obtained using the tight binding model with itinerant bands ($J=0$) at the particular $3/4$ filling. Does this mean that the Fermi surface is obtained from the "non-magnetic" band structure? "Non-magnetic" and "Randomly oriented" are different. They should clarify this.

3. The occupation of Co d bands at the $3/4$ filling seems to be a special point to realize the Van Hove singularity and the triple-Q nesting vectors. However, it is not clear how this particular occupation will be realized in the Co intercalated TaS₂ structure. Is there any simple argument for this?

4. The authors show the comparison of the Fermi surfaces with/without the 3Q order in Fig.3a and Fig.3b. What about the 1Q magnetic order Fermi surface using the same Hamiltonian? The experimental phase diagram shows the 3Q magnetic order changes to the 1Q order as temperature is elevated. Do the authors expect the Fermi surface or the band structure also changes qualitatively at higher temperatures? Do they have the experimental evidence of ARPES measurements?

5. The authors argue that the Fermi surface nesting vectors at the $3/4$ filling are $(\pi,0)$ -G in Fig.3a. Are these vectors consistent with three magnetic ordering Q vectors in the non-coplanar magnetic structure measured experimentally when $x<0.33$?

6. The experimental ARPES data shows that the Co band minimum with the Van Hove singularity shifts toward the Fermi energy upon electron doping. Can this be explained using the tight-binding model they used? Is this change originated from

the reduction of the spin-exchange J or the hopping parameter t , or the chemical potential?

7. It would be worthwhile to check if the change of Ta band minimums due to the doping effect makes sense quantitatively (Fig.2e). I would suggest performing DFT band calculation of TaS₂ and plot the Fermi energy vs the doping level (total number of electrons) with the rigid-band shift approximation. One could check if the slope of the plot is consistent with the Ta band minimum slope in Fig.2e.

8. I think that each Co ion obeys the trigonal crystal field symmetry (D_{3h}) due to the surrounding S ions and, as a result, dxz and dyz orbitals should be degenerate and have higher energies as they are pointing toward S ion directions. At the same time, Co dz² orbitals are covalently bonded with Ta dz² orbitals and they form the bonding/anti-bonding states with partially filled bands near the Fermi energy. Based on the linearly polarized ARPES measurements, the authors argue that the partially filled Co bands are mostly dz² orbitals and dxy/dyz orbitals are located ~0.75eV below the Fermi energy. Is the dxy orbital typo? I think it should be dxz orbital. Overall, the authors should extend their arguments about the orbital characters of measured bands to be consistent with the local symmetry and the crystal field splitting of Co ions. Moreover, they could compare the ARPES measured bands to the orbital-projected DFT band structure obtained from ab-initio calculations.

As a result, I would not recommend the current version of the manuscript to the publication in Nature Communications.

Reviewer #2

(Remarks to the Author)

The manuscript by Luo et al. reports electronic signature of the peculiar 3Q magnetic order in a Van der Waals quantum magnet CoTaS₂ ($x \sim 0.33$). The authors performed high-resolution ARPES measurements on samples with different dopings and revealed an "inverse Mexican hat" dispersion along the K-M-K direction, which was attributed to the van Hove singularity from the 3Q magnetic order. The key evidence for the 3Q magnetic order is the doping dependence of the van Hove singularity, where a change in the intensity ratio between the tip and M points and a variation of the band dispersion near the M point can be identified. In general, the experiments were carried out systematically on samples with different dopings and the experimental data were of good quality. The results are reasonably captured by model calculations. The paper also deals with an interesting topic of a novel magnetic order on a 2D triangular lattice, which can be of broad interest for the community. As far as I know, the paper also presents for the first time the electronic signature of the 3Q magnetic order, which is very timely and well suited for publication in Nature Communications.

While the doping dependence provides electronic signature of the 3Q magnetic order, another key (probably more convincing) evidence should come from the temperature-dependent measurements, which unfortunately are not discussed at all in the current paper. This is my major concern for this paper. Considering that the transition temperatures (~40 K from PM to single Q, ~20 K from single Q to 3Q) are easily accessible for state-of-art ARPES measurements, results from temperature-dependent ARPES measurements are highly desirable and can be very important to confirm the origin. In particular, one would expect that the change of the electronic structure from the 3Q magnetic order can be verified by varying the temperature across the magnetic transition, similar to the change caused by doping. On the other hand, if no obvious temperature-driven change can be observed, the origin of the van Hove singularity might have to be reconsidered. Therefore, I would strongly recommend that the authors present the temperature-dependent ARPES data for samples with 3Q magnetic order and discuss the implications of the results.

Other minor issues:

1. The authors use two different approaches to understand the observed electronic states in CoTaS₂. Namely, the dispersive bands are explained by electron doping of the pristine TaS₂ bands (Fig. 1), while the flat bands near EF (from Co 3d) are compared with model calculations of a 2D triangular lattice (Fig. 4). Strictly speaking, the electronic states should be explained within the same framework, e.g., using DFT calculations for CoTaS₂. I assume that DFT calculations for bulk CoTaS₂ cannot account well for the observed bands. Some explanation on the rationale for such separate treatment of electronic states might be helpful for the readers.
2. Regarding the hole-like delta pocket at the M point: it seems to me that this band might simply be the tail of the gamma pocket centered at K. Why the authors specifically assign it as a separate delta band? Some explanation might be helpful.
3. Page 8, third line from the bottom: "we attribute the observed enlargement of the gamma pocket to the weakening of the 3Q magnetic order above x_c ". This is confusing: the enlargement of the gamma pocket should be simply caused by increased Co doping. Why is it related to the 3Q magnetic order directly?
4. What is the difference between Fig. 3(h) and Fig. 4(b)? They look very similar and appear to be redundant.

Reviewer #3

(Remarks to the Author)

In the article "Discovery of Van Hove Singularities: Electronic Fingerprints of 3Q Magnetic Order in a Van der Waals Quantum Magnet" H.-L. Luo et al. report an indirect evidence of a 3Q magnetic order formation in Co-doped 2H-TaS₂.

Single crystals of pure 2H-TaS₂ and Co-doped TaS₂, with doping levels ranging from 0.29 to 0.36, were investigated using angle-resolved photoemission spectroscopy (ARPES), with pure TaS₂ serving as the reference sample. A critical doping

level of 0.33 was identified, above which the magnetic order in Co-doped TaS₂ transitions from a 3Q antiferromagnetic (AFM) state (at low temperatures) to a coplanar helical AFM order. ARPES measurements reveal indirect signatures of the 3Q magnetic order, through an appearance of new bands whose dispersion cannot be explained in the absence of a 3Q order.

While I find the study interesting and appreciate the careful and detailed experimental work, I have a few questions and concerns that I would like to see addressed before recommending this manuscript for publication.

1. Page 2, lines 37-28: The authors write that “Magnetically intercalated TMD systems mark the emergence of a new frontier: van der Waals magnets”. Does this refer to an emergence of vdW magnets from a historic perspective, as today we have quite a few truly magnetic vdW materials that do not need any intercalation?
2. What is the reason for majority of ARPES measurements to be done at doping of 0.32? This is very close to a critical doping of 0.325 where 3Q order is replaced by helical order.
3. Page 5, lines 128-129: The authors write that the direct comparison of TaS₂ and Co_{0.32}TaS₂ reveals several striking differences, but only 2 are listed (significant electron-doping of α and β bands and appearance of a δ band). While two is technically several, it would be more correct to explicitly write “two striking differences”, except if this refers to also effects (i)-(iv) on page 6.
4. Page 10, growth methods: Are 0.36 samples grown the same way as 0.29, 0.32 and 0.34? 0.36 samples are not listed in the growth methods.
5. On page 10 it says that ARPES measurement temperature was 10 K, however, experimental methods list 7 K. What is the correct measurement temperature?
6. Page 11, ARPES details: What is the reason for performing TaS₂ and Co-doped TaS₂ measurements at different photon energies? Is it related to their matrix elements, or to a different size of the 3D unit cell?
7. In the article authors write that there are 2 different terminations one can end up with after cleaving, TaS₂ one and Co one. Am I correct to assume that the data shown in the main article is from the TaS₂ plane? This should be made more clear in the article.
8. Related to my previous question, do authors observe the same type of behaviour (appearance of the same bands, as well as the same trends with doping) on the Co plane as well?
9. Page 9 in the SI, Figure S8: The authors observe a Mexican hat-like dispersion for samples with 0.29, 0.32 and 0.33 doping, while at 0.34 and 0.36 this turns into a simple hole-like band without a dip at the M point. Can authors comment as to why they see the dip also at 0.33 doping since this is above the critical doping value (0.325 listed in the main text) and one would expect that 0.33 should behave the same way as 0.34 and 0.36

Version 1:

Reviewer comments:

Reviewer #1

(Remarks to the Author)

I went through the authors' response letter, and I think they addressed my questions/concerns properly. Therefore, I can recommend the publication of this manuscript.

Reviewer #2

(Remarks to the Author)

I thank the authors for the efforts to answer all the relevant questions. Although I find that most of issues are well addressed, I am still not convinced that the new data in Fig. S14 provides “additional” support for the electronic signature of the 3Q magnetic phase. Specifically, the temperature-dependent change in the EDC of Fig. S14(d-f) is very subtle and can be easily affected by the thermal broadening, i.e., the Mexican-hat bands might still be present in the non-magnetic phase, but it becomes invisible due to thermal broadening at elevated temperatures. In general, the temperature-induced change in the experimental band structure is very small compared to theoretical predictions. Some clarification on this point will certainly be important to a comprehensive understanding of the possible 3Q magnetic phase.

Version 2:

Reviewer comments:

Reviewer #2

(Remarks to the Author)

I thank the authors for addressing this question. The detailed simulation now provides a compelling evidence for the electronic transition associated with the triple-Q phase. Therefore, I recommend the publication of this paper in Nature Communications in the current form.

Response to Reviewer's Comments:

We thank all reviewers for the careful reviewing of our paper and his/her constructive comments and suggestions to improve the manuscript. We also thank the reviewer for nicely capturing the importance and significance of our work.

Response to Reviewer #1:

This manuscript reports the angle resolved photoemission spectroscopy (ARPES) study of Co_xTaS_2 providing the evidence of the electronic structure to support the sharp magnetic transition from a non-coplanar 3Q order to a helical order upon electron doping at low temperature. Co_xTaS_2 forms an intercalated Co triangular lattice between TaS₂ layers and the doping of Co ions can change the electronic structure dramatically near $x=0.33$, possibly accompanied by the magnetic transition. The main conclusion of this manuscript is that the electron doping effect mainly changes Ta band minimum as evidenced from the ARPES measurements of both Co dopings and K depositions, while the changes of Co-derived bands are evident from only the Co doping effect which is attributed to the spin-exchange interaction change of itinerant bands with localized spins. The manuscript is clearly written and address an important and interesting topic in these Van-der-Waals quantum magnets. While these findings can be of great interests to other similar intercalated transition metal dichalcogenides, I am not convinced that a rather simple tight-binding model that they used can explain the essential physics of Co-derived bands near the Fermi energy and the parameters they used are not justified.

(1) Reviewer's comment: To understand the ARPES band changes upon electron dopings, the authors used a rather simple tight-binding model including the nearest-neighbor hopping parameter t and the Kondo coupling J between itinerant electrons and the classical spins. However, there is no justification of the parameters they used. In my opinion, it would be beneficial to use some ab-initio calculations to verify the quantitative features of band structure changes near the Fermi energy. For example, the authors should at least check if the itinerant Co d band ($J=0$) is consistent with the Co projected DFT band structure of $\text{Co}_{1/3}\text{TaS}_2$. Moreover, these itinerant Co d_{z^2} bands are covalently bonded with Ta d_{z^2} bands. They should justify why these Ta d_{z^2} orbitals can be neglected in this simple tight-binding model. They should also check how sensitively the band structure near the M point changes due to the change of t and J parameters.

Our response: We thank the referee for this insightful comment. Indeed, we employ a simple tight-binding model that couples to local magnetic moments to describe the flat Co-derived bands near the Fermi level. The model parameters, t and J , are not arbitrary; they are fitted to

experimental data. In our view, the ability of such a minimal model to capture the salient near- E_f features, such as the inverse Mexican-hat dispersion observed in ARPES, is quite remarkable and worth reporting. Furthermore, the simplicity of the model highlights the essential interplay between the itinerant electrons and the local moments – an aspect that is often difficult to isolate in more complex approaches. As we discuss below, DFT sometimes encounters difficulties in dealing with particularly flat-band features, primarily due to strong electron-electron correlations arising from the flatness and the presence of local magnetic moments. Under these circumstances, we believe it is reasonable to employ a phenomenological model to account for the flat-band physics that DFT misses.

To address the referee's concern, we performed orbital-projected DFT calculations for $\text{Co}_{1/3}\text{TaS}_2$ without and with the 3Q magnetic order and compared the results with our ARPES data. Figures R1(a, b) and R1(d, e) show the bulk band structures for the non-magnetic and 3Q-ordered states, respectively. The highly-dispersive α and β bands and the near- E_f flat-bands (γ and δ bands) are qualitatively reproduced. Moreover, by comparing the band structures projected onto the Co $3d$ (Fig. R1(a, d)) and Ta $5d$ (Fig. R1(b, e)) orbitals, we find that the near- E_f flat bands are primarily derived from Co $3d$ orbitals, while the α and β bands mainly originate from Ta $5d$ orbitals, which justifies neglecting Ta $5d$ orbitals in our tight-binding model.

Nevertheless, there are several reasons for not relying solely on DFT to describe our results. First, introducing the 3Q magnetic order (Fig. R1(d, e)) produces results that deviate notably from the non-magnetic calculation (Fig. R1(a, b)). Such sensitivity of the band structure to the presence/absence of 3Q magnetic order is not observed experimentally. Second, DFT struggles to accurately reproduce the detailed dispersions of the experimental γ and δ bands, likely due to strong correlation effects associated with their flatness and the influence of local magnetic moments. Third, DFT results are quite sensitive to the computational setup. For instance, previous reports [1, 2] show qualitative agreement with ours but differ in quantitative details, suggesting that even relatively small changes in parameter settings may easily obscure shallow γ and δ features observed experimentally. Finally, the band-unfolding process introduces additional modifications to the calculated band structures, which further complicate the identification of individual bands (see Fig. R1f). Given these limitations of DFT and the distinct orbital characters of the highly-dispersive and near- E_f flat bands, we consider it is more appropriate to employ a phenomenological tight-binding model on a Co triangular lattice. This model effectively captures the essential flat-band physics and its coupling to local moments that DFT cannot describe with sufficient accuracy.

In the revised manuscript, we have added Fig. R1 as Supplementary Fig. S5, and the corresponding *DFT Calculations* section in *Methods*. In addition, we included a discussion on the orbital characters (page 6, line 162): “We further performed DFT calculations for $\text{Co}_{1/3}\text{TaS}_2$ and identified highly-dispersive bands mainly originating from Ta orbitals together with near- E_f flat bands

primarily deriving from Co orbitals (see Supplementary Fig. S5).” Moreover, we have discussed the feasibility of using a tight-binding model on a Co triangular lattice (page 8, line 220): “Since DFT struggles to accurately describe the dispersion of the near- E_f flat bands, primarily due to strong electron-electron correlations and the presence of local magnetic moments, it cannot provide a reliable account of our system. At the same time, the near- E_f flat bands are dominated by Co $3d$ orbitals, whereas the dispersive α and β bands mainly originate from TaS₂-derived states (see Supplementary Fig. S5). Given both the limitations of DFT and the distinct orbital characters of these states, we adopt a phenomenological tight-binding model on a Co triangular lattice to account for the flat bands and their coupling to local moments.”

Fig. R1. DFT band structures of $\text{Co}_{1/3}\text{TaS}_2$ without and with 3Q magnetic order. **(a, b)** Calculated bulk band structures of $\text{Co}_{1/3}\text{TaS}_2$ without 3Q magnetic order, projected onto the Co $3d$ (a) and Ta $5d$ (b) orbitals. **(c)** Brillouin zones of 2H-TaS₂ (solid hexagon), $\text{Co}_{1/3}\text{TaS}_2$ without 3Q order (black dashed hexagon), and $\text{Co}_{1/3}\text{TaS}_2$ with 3Q order (red dashed hexagon). **(d, e)** Calculated bulk band structures of $\text{Co}_{1/3}\text{TaS}_2$ with 3Q magnetic order, projected onto the Co $3d$ (d) and Ta $5d$ (e) orbitals. **(f)** DFT band structure of $\text{Co}_{1/3}\text{TaS}_2$ with 3Q magnetic order unfolded to the Brillouin zone of the non-magnetic $\text{Co}_{1/3}\text{TaS}_2$.

In Fig. R2, we study the sensitivity of the band dispersions near the M point to variations of the parameters t and J within our tight-binding model. Specifically, we calculated the band structures for different values of J at fixed t , and for different values of t at fixed J . We find that a finite exchange coupling J can induce the emergence of an inverse Mexican hat-like dispersion around the M point. Meanwhile, changing the magnitudes of J and t primarily affects the bandwidth and the energy separation between the M-point minimum and the neighboring maxima. Therefore,

the inverse Mexican hat-like feature is qualitatively preserved as long as the 3Q magnetic order is present. In the main text, we adopt the parameter set that best reproduces the experimental observations for comparison.

In the revised manuscript, we have included Fig. R2 as Supplementary Fig. S12 to demonstrate the preservation of the inverse Mexican hat-like band in the presence of 3Q order. In addition, we added the following sentence on page 10, line 283: “We further tested the robustness of this result by varying hopping amplitude t and coupling constant J , and found that the inverse Mexican hat-like dispersion persists as long as the 3Q order is present (See Supplementary Fig. S12).”

Fig. R2. Sensitivity of the band structures near the M point to variations of the hopping amplitude t and exchange coupling J . **(a)** Band structures calculated for different values of t at a fixed $J = 0.04$. **(b)** Band structures calculated for different values of J at a fixed $t = -0.062$. All calculations were performed with the chemical potential $\mu = 2.545t$.

(2) Reviewer’s comment: The authors argue that Fig.3a Fermi surface is obtained from the absence of the 3Q magnetic order- “namely, when the magnetic moments are randomly oriented”. However, it looks like the Fermi surface can be obtained using the tight binding model with itinerant bands ($J=0$) at the particular $\frac{3}{4}$ filling. Does this mean that the Fermi surface is obtained from the “non-magnetic” band structure? “Non-magnetic” and “Randomly oriented” are different. They should clarify this.

Our response: We thank the reviewer for the correction. We agree that describing the Co moments as “randomly oriented” in the paramagnetic state is misleading. An accurate term should be “non-magnetic”, which reflects the absence of long-range magnetic order, rather than implying the presence of fluctuating local moments with random orientations. To avoid confusion, we have revised the wording in the manuscript accordingly. Specifically, the sentence “namely, when the magnetic moments are randomly oriented” has been changed to “that is, in the non-magnetic state (exchange coupling constant $J=0$)” on page 8, line 228.

(3) Reviewer’s comment: The occupation of Co d bands at the $\frac{3}{4}$ filling seems to be a special point to realize the Van Hove singularity and the triple- Q nesting vectors. However, it is not clear how this particular occupation will be realized in the Co intercalated TaS₂ structure. Is there any simple argument for this?

Our response: We thank the referee for this insightful comment. According to a simple electron-counting scheme, the Co-intercalated compounds can be regarded as alternating layers of $[\text{Co}_{1/3}]^+$ and $[\text{TaS}_2]^-$ (Fig. R3(a)). As shown in Fig. R3(b), the coordination environment of Co^{3+} in $\text{Co}_{1/3}\text{TaS}_2$ is a trigonally distorted pseudo-octahedron with D_{3d} crystal field symmetry, leading to a splitting of the Co $3d$ orbitals into e_2 ($d_{xy}, d_{x^2-y^2}$), a_1 (d_{z^2}), and e_1 (d_{xz}, d_{yz}) (Fig. R1(c)) [3, 4].

Polarization-dependent ARPES measurements (Figure 1 of our manuscript) suggest that the shallow γ and δ bands near the Fermi level are most likely dominated by Co d_{z^2} character. We therefore focus on the filling of the d_{z^2} orbital. For a Co^{3+} ion with six d electrons within a D_{3d} crystal field, if the crystal field from the TaS₂ layers were very weak, the electrons would adopt a high-spin configuration as sketched in the left panel of Fig. R3(c), with $n = 4$ unpaired electrons. In this case, the d_{z^2} orbital would be only half-filled (1/2-filling). However, such a configuration would yield an effective magnetic moment of $\mu_{\text{eff}}(\mu_B) = \sqrt{n(n+2)} \approx 4.9$, which is noticeably larger than the experimental value of $\mu_{\text{eff}}(\mu_B) = 3.7$ obtained from magnetic susceptibility measurements [5]. This suggests that the crystal field may be not as weak as in the pure high-spin limit, and that an intermediate spin configuration is stabilized by the crystal field and covalency. In this situation, more electrons are transferred from the d_{xz}/d_{yz} orbitals into the d_{z^2} and $d_{xy}/d_{x^2-y^2}$ states, effectively increasing the d_{z^2} occupancy from the 1/2-filling expected in the high-spin limit toward $\sim 3/4$ filling, as illustrated in the right panel of Fig. R3(c).

While this picture is simplified, it provides a sound rationale for the $3/4$ -filling condition of the Co lattice in $\text{Co}_{1/3}\text{TaS}_2$, consistent with the resemblance between the ARPES-observed γ Fermi surface and the tight-binding prediction for a $3/4$ -filled Co band (see Figure 3 in our manuscript).

Fig. R3. (a) Crystal structure of $\text{Co}_{1/3}\text{TaS}_2$, illustrating the formal charges of the TaS_2 and Co layers on a simple electron-counting scheme [4]. (b) Local coordination of a Co center with surrounding S atoms in $\text{Co}_{1/3}\text{TaS}_2$, forming a D_{3d} ligand field [4]. (c) Schematic d-orbital splitting diagrams for Co^{3+} ($3d^6$) in a D_{3d} ligand field. Left panel: high-spin configuration; Right panel: relatively lower-spin configuration.

(4) Reviewer's comment: The authors show the comparison of the Fermi surfaces with/without the 3Q order in Fig.3a and Fig.3b. What about the 1Q magnetic order Fermi surface using the same Hamiltonian? The experimental phase diagram shows the 3Q magnetic order changes to the 1Q order as temperature is elevated. Do the authors expect the Fermi surface or the band structure also changes qualitatively at higher temperatures? Do they have the experimental evidence of ARPES measurements?

Our response: We thank the referee for this insightful comment. To address this point, we have simulated the band structure of the 1Q ordered state using a tight-binding model. Figure R4(a) shows the 1Q ordered structure with two adjacent layers, while Fig. R4(b) illustrates the Brillouin zones of the unit cell ($a \times a$) without magnetic order and of the supercell ($2a \times a$) with 1Q order. The band structures are modeled by the following Hamiltonian defined on the $2a \times a$ lattice:

$$H = -\sum_{\langle ij \rangle_{1,\alpha}} t_1 c_{i\alpha}^\dagger c_{j\alpha} - \sum_{\langle ij \rangle_{2,\alpha}} t_2 c_{i\alpha}^\dagger c_{j\alpha} - \sum_{\langle ij \rangle_{3,\alpha}} t_3 c_{i\alpha}^\dagger c_{j\alpha} - \mu \sum_{i,\alpha} c_{i\alpha}^\dagger c_{i\alpha} - J \sum_i \mathbf{S}_i \cdot$$

$$c_{i\alpha}^\dagger \boldsymbol{\sigma}_{\alpha\beta} c_{i\beta}.$$

We tested a wide range of parameter sets for t_1 , t_2 , and t_3 to reproduce the overall hole-like band along the K-M-K' direction, and then examined the effect of varying the exchange coupling constant J . In Fig. R4(c), we show a few representative cases. Our calculations consistently reveal

that, regardless of whether $J=0$ (without 1Q magnetic order) or $J \neq 0$ (with 1Q magnetic order), the band dispersion along K-M-K' remains hole-like. This indicates that the presence or absence of the 1Q magnetic order does not qualitatively alter the dispersion along this path.

Fig. R4. Band structure of the 1Q ordered state. **(a)** Schematic illustration of the 1Q ordered structure with two adjacent layers. Large (small) dots denote sites in the upper (lower) layer. Blue/red colors indicate opposite out-of-plane magnetic-moment orientations. The hopping parameters t_1 , t_2 , and t_3 are assigned according to the bond lengths, and the supercell ($2a \times a$) is shown in comparison with the unit cell ($a \times a$). **(b)** Brillouin zones of the unit cell ($a \times a$) without 1Q order and of the supercell ($2a \times a$) with 1Q order, showing the high-symmetry K-M-K' path. **(c)** Band dispersions calculated from the tight-binding Hamiltonian on the $2a \times a$ lattice for different values of J with fixed $t_1=0.08$, $t_2=-0.04$, $t_3=-0.005$, and $\mu=-2.85t_1$.

To address the referee's question on temperature dependence, we performed temperature-dependent ARPES measurements on $\text{Co}_{0.31}\text{TaS}_2$ along K-M-K (Fig. R5). As shown in Fig. R5(a), at 6 K within the 3Q ordered state, the near- E_F band exhibits an inverse Mexican hat-like dispersion. The energy distribution curves (EDCs) obtained at M (k_1) and k_2 [Fig. R5(d)] show that the peak position of EDC(k_1) is located at a slightly yet clearly higher binding energy than that of EDC(k_2). This behavior is consistent with a 3Q order-induced reconstruction that produces the inverse Mexican hat-like dispersion described in our manuscript.

By contrast, at 31 K (1Q order) and 82 K (paramagnetic state), the dispersions in Fig. R5(b, c) no longer exhibit the inverse Mexican hat-like structure. Instead, the EDC peak positions at k_1 and k_2 [Fig. R5(e, f)] are nearly identical, indicating a hole-like band dispersion along K-M-K.

Taken together, these results demonstrate that the inverse Mexican hat-like dispersion emerges exclusively in the 3Q phase, while the 1Q and paramagnetic phases show only a hole-like band. This provides direct spectroscopic evidence that the 3Q magnetic order strongly modifies the low-energy electronic structure.

In the revised manuscript, we have added Fig. R5 as Supplementary Fig. S14, Fig. R4 as Supplementary Fig. S15, and the corresponding *Theory* section in *Methods*. In addition, we have expanded the discussion on page 11, line 304 to clarify this point: “Temperature-dependent measurements provide further evidence for the role of 3Q magnetic order. As shown in Supplementary Fig. S14, the inverse Mexican hat-like dispersion is observed only in the 3Q ordered phase at low temperature. At higher temperatures, corresponding to the 1Q ordered and paramagnetic states, the dispersion evolves into a simple hole-like band along the K-M-K direction. These observations are in close agreement with tight-binding calculations, which shows that only the 3Q ordered state gives rise to the inverse Mexican hat-like dispersion (Fig. 4 and Supplementary Fig. S15). Together, these results demonstrate that the reconstruction of the near- E_F bands is a distinctive fingerprint of the 3Q order.”

Fig. R5. Temperature-dependent ARPES spectra of $\text{Co}_{0.31}\text{TaS}_2$ along the K-M-K direction. **(a-c)** Band structures measured at 6K, 31K and 82K, respectively. Arrows mark the momenta k_1 (M point) and k_2 . **(d-f)** EDCs extracted at k_1 and k_2 from panels (a-c).

(5) Reviewer’s comment: The authors argue that the Fermi surface nesting vectors at the $\frac{3}{4}$ filling are $(\pi, 0)$ -G in Fig.3a. Are these vectors consistent with three magnetic ordering Q vectors in the

non-coplanar magnetic structure measured experimentally when $x < 0.33$?

Our response: We thank the referee for this helpful question. We clarify at the outset that the Fermi-surface nesting vectors $(\pi, 0)$ -G at $3/4$ filling correspond directly to the three magnetic ordering Q vectors. As shown in Supplementary Fig. S9, which we reproduced here as Fig. R6, the 3Q magnetic order in samples with $x < 0.33$ leads to a 2×2 magnetic unit cell in real space (Fig. R6(b)), relative to the original Co triangular lattice (Fig. R6(a)). The original lattice has two basis vectors with equal length a separated by 60° , and the 2×2 magnetic superstructure thus doubles the periodicity along both directions.

Accordingly, the periodicity in reciprocal space is reduced (Fig. R6(c)). In the original lattice, the real-space periodicity a corresponds to a reciprocal lattice periodicity of $2\pi/a$, which defines the extent of the original Brillouin zone. The 2×2 enlargement in real space leads to a folding of the Brillouin zone, and new magnetic ordering wave vectors \mathbf{Q} of magnitude $2\pi/2a (= \pi/a)$ emerge along three high-symmetry directions separated by 120° . These Q vectors are $Q_1 = (\pi, 0)$, $Q_2 = (-\pi/2, \sqrt{3}\pi/2)$, and $Q_3 = (-\pi/2, -\sqrt{3}\pi/2)$, expressed in units of $1/a$.

Finally, the nesting vector labeled as $(\pi, 0)$ - G in Fig. 3a, which connects the edge centers of the hexagonal Fermi surface at $3/4$ filling, is equivalent to Q_1 up to a reciprocal lattice vector. This supports our interpretation that the observed Fermi surface reconstruction originates from the 3Q magnetic order, consistent with our discussion in the manuscript.

Fig. R6. The 3Q magnetic order in real space and the reconstructed Brillouin zone. **(a)** The triangular lattice of intercalated Co atoms (indicated by gray dashed lines) and its unit cell (outlined by black lines). **(b)** Configuration of the 3Q magnetic order with four sublattices and the resultant 2×2 magnetic unit cell (black lines). **(c)** The Brillouin zone of the Co triangular lattice, shown both without (gray dashed lines) and with (black solid lines) the 3Q magnetic order. The arrow indicates the folding wave vector $(\pi, 0)$. **(d)** Nesting wave vector $(\pi, 0)$ -G that connects the edge centers of the hexagonal Fermi surface at $3/4$ -filling. The nesting vector $(\pi, 0)$ -G is equivalent to $(\pi, 0)$

up to a reciprocal lattice vector G .

(6) Reviewer's comment: The experimental ARPES data shows that the Co band minimum with the Van Hove singularity shifts toward the Fermi energy upon electron doping. Can this be explained using the tight-binding model they used? Is this change originated from the reduction of the spin-exchange J or the hopping parameter t , or the chemical potential?

Our response: We thank the referee for the careful observation. It is indeed true that the Co band minimum shifts toward the Fermi level as the Co composition increases from 0.34 to 0.36. However, this behavior cannot be fully captured within our tight-binding model. As shown in Fig. R2(a), reducing J would broaden the bandwidth, which is opposite to the experimentally observed upward shift of the Co band minimum. Moreover, although a reduction in t (Fig. R2(b)) could in principle narrow the bandwidth and shift the band minimum upward, we do not find any physical reason for t to decrease with increasing Co content. Lastly, a rigid-band picture governed by the chemical potential also fails to explain the observation, since μ remains experimentally pinned near the band top and cannot account for the upward shift. To reasonably explain the data, we propose that the helical antiferromagnetic order emerging for $x > 0.33$ plays a dominant role, strongly modifying the electronic structure and pushing the Co band bottom upward. However, the complexity of this order along the b^* - c plane lies beyond the scope of our tight-binding model.

(7) Reviewer's comment: It would be worthwhile to check if the change of Ta band minimums due to the doping effect makes sense quantitatively (Fig.2e). I would suggest performing DFT band calculation of TaS₂ and plot the Fermi energy vs the doping level (total number of electrons) with the rigid-band shift approximation. One could check if the slope of the plot is consistent with the Ta band minimum slope in Fig.2e.

Our response: We thank the referee for this constructive suggestion. Following the comment, we performed DFT calculations for 2H-TaS₂ along the M_0 - Γ_0 - K_0 - M_0 direction and obtained the corresponding density of states (DOS) integrated over the entire momentum space, as shown in Fig. R7a. From this DOS, we determined the relation between the Fermi energy and the doping level, expressed as the number of additional electrons per TaS₂ unit cell (Fig. R7b).

To facilitate a direct comparison with experiment, we estimated that each Co atom in Co x TaS₂ nominally dopes approximately $3x$ electrons to the TaS₂ layers, giving a total electron count of $3x$ per unit cell. We then plotted the evolution of the α -band minimum along the Γ_0 - M_0 direction as a function of this nominal electron doping (black curve in Fig. R7c, reproduced from Fig. 2e of our manuscript). When comparing this experimental trend with the calculated dependence of Fermi-level shift on electron doping (blue curve in Fig. R7c), we find excellent agreement not only

in the absolute electron count but also in the slope of the relation between the number of electrons and the Fermi-level shift.

This quantitative consistency strongly supports that the α and β bands are predominantly derived from TaS₂ orbitals, and that their energy shifts upon electron doping are well captured by the rigid-band shift approximation. To substantiate this point, we have added Fig. R7 as Supplementary Fig. S4 and incorporated the following sentence into the manuscript (page 6, line 149): “Density Functional Theory (DFT) calculations for 2H-TaS₂ reveal that the observed $\sim 300\text{meV}$ band shift can be reproduced by adding one electron per TaS₂ unit cell, in good agreement with a simple electron-counting picture of Co_{1/3}TaS₂ consisting of alternating [Co_{1/3}]⁺ and [TaS₂]⁻ layers (see Supplementary Fig. S4).” In addition, we have added another sentence on page 7, line 189: “The observed slope of the shift of the α -band minimum shift as a function of Co doping is well reproduced by a rigid-band-shift approximation applied to the TaS₂ band structure (see Supplementary Fig. S4)”

Fig. R7. Calculated and experimental electron-doping dependence of TaS₂ bands. **(a)** Calculated band structure of 2H-TaS₂ along the M_0 - Γ_0 - K_0 - M_0 direction. The right panel displays the calculated density of states (DOS) integrated over the entire Brillouin zone as a function of energy. **(b)** Calculated relation between the number of additional electrons per TaS₂ unit cell and the upward shift of the Fermi level. **(c)** Comparison between experiment and calculation: the black curve represents the evolution of the α -band minimum along the Γ_0 - M_0 direction as a function of the nominal electrons per TaS₂ unit cell ($3x$, taken from Fig. 2e in the manuscript), while the blue curve shows the calculated relation between electron doping and Fermi-level shift.

(8) Reviewer’s comment: I think that each Co ion obeys the trigonal crystal field symmetry (D3h) due to the surrounding S ions and, as a result, dxz and dyz orbitals should be degenerate and have higher energies as they are pointing toward S ion directions. At the same time, Co dz^2

orbitals are covalently bonded with Ta dz_2 orbitals and they form the bonding/anti-bonding states with partially filled bands near the Fermi energy. Based on the linearly polarized ARPES measurements, the authors argue that the partially filled Co bands are mostly dz_2 orbitals and dxy/dyz orbitals are located $\sim 0.75\text{eV}$ below the Fermi energy. Is the dxy orbital typo? I think it should be dxz orbital. Overall, the authors should extend their arguments about the orbital characters of measured bands to be consistent with the local symmetry and the crystal field splitting of Co ions. Moreover, they could compare the ARPES measured bands to the orbital-projected DFT band structure obtained from ab-initio calculations.

Our response: We thank the referee for these insightful comments. As explained in the response to Comment #3, the coordination environment of Co^{3+} in $\text{Co}_{1/3}\text{TaS}_2$ is a trigonally distorted pseudo-octahedron (D_{3d} crystal field symmetry), leading to a qualitative d -orbital splitting diagram of $e_2(dxy, dx^2-y^2)$, $a_1(dz_2)$, and $e_1(dxz, dyz)$. For simplicity, we treat Co^{3+} in the high-spin d^6 configuration as shown in the left panel of Fig. R8(b). In contrast, Ta^{3+} has D_{3h} symmetry, giving the d -orbital splitting: $a_1(dz_2)$, $e_2(dxy, dx^2-y^2)$, and $e_1(dxz, dyz)$ ^[4,5], with a d^2 configuration as shown in the right panel of Fig. R8(b).

We then consider what determines the orbital spectral weight in ARPES. The spectral weight is strongly modulated by the dipole matrix element $|\mathbf{M}_{f,i}^k|^2 \propto |\langle \phi_f^k | \hat{\mathbf{e}} \cdot \mathbf{r} | \phi_i^k \rangle|^2$, where ϕ_i^k (ϕ_f^k) denotes the initial (final) state wavefunction of the photoelectron, and $\hat{\mathbf{e}}$ is the unit vector along the polarization direction of the incident light (LH: even parity; LV: odd parity). Approximately, only orbitals sharing the same parity as the incident polarization can be detected in ARPES [6]. When the analyzer slit is aligned along the Γ_0-K_0 direction (parallel to the x -axis as defined in Fig. R8(c)), the d orbital symmetries of Co and Ta (a_1 , e_1 , and e_2 sets) with respect to the scattering plane (x - z plane) are illustrated in Fig. R8(d). The even dz_2 orbitals are expected to be visible with LH polarization, whereas the $e_1(dxz, dyz)$ and $e_2(dxy, dx^2-y^2)$ sets are not symmetric collectively with respect to the scattering plane and may appear under both LH and LV polarizations. Accordingly, the observed bands can be classified into three categories: (1) the γ_K and γ_M bands, visible only under LH polarization and thus mainly derived from dz_2 orbitals; (2) the α and β bands, visible under both LH and LV polarizations but significantly stronger in LH, which can arise from all orbitals but are dominated by dz_2 character; and (3) the ε band, which appears predominantly under LV polarization is mainly contributed by dxz/dyz and dxy/dx^2-y^2 sets.

Together with the DFT-calculated band structures projected onto individual Co and Ta orbitals (see Fig. R9), the above polarization-dependent analysis can be further extended. The DFT results enable a clear separation of Co- and Ta-derived contributions, indicating that the near- E_f γ_K and γ_M bands mainly arise from Co $3dz_2$ orbital; whereas the highly-dispersive α and β bands are dominated by Ta $5dz_2$ and Ta $5dxy/dx^2-y^2$ states.

To strength our argument, we have added the phrase “a pronounced reduction of spectral weight in the highly-dispersive α and β bands” on page 6, line 170, and included “and the DFT-calculated orbital projected band structures (see Supplementary Note and Supplementary Fig. S6-S7)” on page 6, line 172. In addition, we have revised the sentence on page 7, line 174, to read: “the near-Ef γ_K and γ_M bands mainly originate from Co $3d_{z^2}$ orbital, the highly-dispersive α and β bands are dominated by Ta $5d_{z^2}$ Ta $5d_{xy}/d_{x^2-y^2}$ orbitals, and the ε band is primarily contributed by d_{xz}/d_{yz} and $d_{xy}/d_{x^2-y^2}$ states.” We have also added Fig. R8 as Supplementary Fig. S6, Fig. R9 as Supplementary Fig. S7, and included the corresponding discussion as a Supplementary Note in the Supplementary Materials.

Fig. R8. Orbital Symmetries of $\text{Co}_{1/3}\text{TaS}_2$. **(a)** Local coordination environments of Co and Ta centers and their surrounding S atoms in $\text{Co}_{1/3}\text{TaS}_2$. **(b)** Qualitative diagrams of d -orbital splitting for isolated Co and Ta centers under their respective local ligand field. **(c)** Real-space projections on the x - y plane showing the local coordination geometries of Co and Ta. **(d)** Symmetries of the d -orbitals for Co and Ta with respect to the scattering plane, defined as the x - z plane aligned along the Γ_0 - K_0 direction [4].

Fig. R9. DFT-calculated orbital-projected band structures of bulk $\text{Co}_{1/3}\text{TaS}_2$ with 3Q magnetic order. **(a-c)** Calculated band structures of $\text{Co}_{1/3}\text{TaS}_2$ with 3Q order projected onto Co dz^2 , Co dxz/dyz , Co dxy/dx^2-y^2 orbitals, respectively. **(d-f)** Corresponding orbital projections onto Ta dz^2 , Ta dxz/dyz , Ta dxy/dx^2-y^2 orbitals, respectively. The color scale represents the orbital-weight intensity.

As a result, I would not recommend the current version of the manuscript to the publication in Nature Communications.

We sincerely thank the Reviewer for their careful evaluation and constructive feedback. We hope that the revised version has adequately addressed their concerns and that they may now find it suitable for publication in Nature Communications.

Response to Reviewer #2:

The manuscript by Luo et al. reports electronic signature of the peculiar 3Q magnetic order in a Van der Waals quantum magnet CoTaS_2 ($x \sim 0.33$). The authors performed high-resolution ARPES measurements on samples with different dopings and revealed an “inverse Mexican hat” dispersion along the K-M-K direction, which was attributed to the van Hove singularity from the 3Q magnetic order. The key evidence for the 3Q magnetic order is the doping dependence of the van Hove singularity, where a change in the intensity ratio between the tip and M points and a variation of the band dispersion near the M point can be identified. In general, the experiments

were carried out systematically on samples with different dopings and the experimental data were of good quality. The results are reasonably captured by model calculations. The paper also deals with an interesting topic of a novel magnetic order on a 2D triangular lattice, which can be of broad interest for the community. As far as I know, the paper also presents for the first time the electronic signature of the 3Q magnetic order, which is very timely and well suited for publication in Nature Communications.

(1) Reviewer's comment: While the doping dependence provides electronic signature of the 3Q magnetic order, another key (probably more convincing) evidence should come from the temperature-dependent measurements, which unfortunately are not discussed at all in the current paper. This is my major concern for this paper. Considering that the transition temperatures (~ 40 K from PM to single Q, ~ 20 K from single Q to 3Q) are easily accessible for state-of-art ARPES measurements, results from temperature-dependent ARPES measurements are highly desirable and can be very important to confirm the origin. In particular, one would expect that the change of the electronic structure from the 3Q magnetic order can be verified by varying the temperature across the magnetic transition, similar to the change caused by doping. On the other hand, if no obvious temperature-driven change can be observed, the origin of the van Hove singularity might have to be reconsidered. Therefore, I would strongly recommend that the authors present the temperature-dependent ARPES data for samples with 3Q magnetic order and discuss the implications of the results.

Our response: We thank the reviewer for raising this important point, which we also consider crucial for establishing the role of the 3Q magnetic order. To address the reviewer's concern, we performed temperature-dependent ARPES measurements on $\text{Co}_{0.31}\text{TaS}_2$ along the K-M-K direction (Fig. R5). These results clearly demonstrate the temperature-driven evolution of the band structure, consistent with the change from the 3Q to 1Q magnetic order. A detailed description and discussion of these results are provided in our response to Comment#4 of Referee#1, where we compare the temperature-dependent data with tight-binding calculations and discuss distinctive role of the 3Q order.

(2) Reviewer's comment: The authors use two different approaches to understand the observed electronic states in CoTaS_2 . Namely, the dispersive bands are explained by electron doping of the pristine TaS_2 bands (Fig. 1), while the flat bands near EF (from Co 3d) are compared with model calculations of a 2D triangular lattice (Fig. 4). Strictly speaking, the electronic states should be explained within the same framework, e.g., using DFT calculations for CoTaS_2 . I assume that DFT calculations for bulk CoTaS_2 cannot account well for the observed bands. Some explanation on the rationale for such separate treatment of electronic states might be helpful for the readers.

Our response: We thank the referee for raising this important point. We agree that, ideally, both

the dispersive and flat bands should be treated within a unified theoretical framework. However, it is well known that density functional theory (DFT) struggles to accurately describe flat bands and local magnetic moments, primarily due to the strong electron-electron correlations involved (see Fig. R1). To further justify this separation, we note that the near-E_f flat bands are dominated by Co 3*d* orbitals, whereas the dispersive α and β bands are primarily derived from Ta 5*d* states (see Fig. R1). Since the orbital characters of these states are quite distinct, in the absence of a fully first-principles description of all bands, we adopt a hybrid approach: a phenomenological tight-binding model on a Co triangular lattice is employed to describe the flat bands and their coupling to local moments, while a rigid-band-shifted DFT of pristine TaS₂ is used to describe the more dispersive α and β bands (see Fig. R7). This combined framework yields a good quantitative agreement with the ARPES data.

To help readers better understand the rationale behind our separate treatments of the dispersive and flat-band states, we have clarified this point in the revised manuscript (page 8, line 219): “Since DFT struggles to accurately describe the dispersion of the near-E_f flat bands, primarily due to strong electron-electron correlations and the presence of local moments, it cannot provide a reliable account of our system. At the same time, the near-E_f flat bands are dominated by Co 3*d* orbitals, whereas the dispersive α and β bands mainly originate from TaS₂-derived states (see Supplementary Fig. S5). Given both the limitations of DFT and the distinct orbital characters of these states, we adopt a phenomenological tight-binding model on a Co triangular lattice to account for the flat bands and their coupling to local magnetic moments.”

(3) Reviewer’s comment: Regarding the hole-like delta pocket at the M point: it seems to me that this band might simply be the tail of the gamma pocket centered at K. Why the authors specifically assign it as a separate delta band? Some explanation might be helpful.

Our response: We agree with the reviewer that the δ band at the M point and the γ pocket centered at K originate from the same band, exhibiting anisotropic dispersions along different high-symmetry directions, similar to how the α and β bands behave along both Γ -M and Γ -K directions.

However, since we present more detailed studies of these two branches in later sections (e.g., doping-dependent experiments), we find it clearer to distinguish them by direction for subsequent discussion. Therefore, we have renamed the current γ band as γ_K band (along Γ -K) and the previous δ band as γ_M band (along Γ -M), to indicate that they are directional branches of the same band. Consequently, the previous labels for the ϵ and ζ bands are shifted forward by one Greek letter and are now labeled δ and ϵ , respectively. This relabeling is purely for clarity and does not affect our conclusions.

(4) Reviewer's comment: Page 8, third line from the bottom: "we attribute the observed enlargement of the gamma pocket to the weakening of the 3Q magnetic order above x_c ". This is confusing: the enlargement of the gamma pocket should be simply caused by increased Co doping. Why is it related to the 3Q magnetic order directly?

Our response: We thank the reviewer for this thoughtful comment. While increased Co doping could, in principle, lead to band filling, which would simultaneously cause a downward shift of the band bottom and a corresponding enlargement of the γ pocket, our data do not support this expectation. Specifically, as shown in Fig. 2(e), the bottom of the γ_K band does not shift downward in energy with increasing doping. This behavior contradicts the expectation from a simple band-filling scenario.

Therefore, the observed enlargement of the γ pocket cannot be solely attributed to charge doping. A plausible explanation is the weakening of the 3Q magnetic order above $x_c \approx 0.33$, which alters the Fermi surface topology and could lead to an enlarged γ Fermi pocket size, as seen when comparing Fig. 3a to Fig. 3b in our manuscript. Thus, our interpretation goes beyond a simple doping picture and highlights the essential role of the 3Q magnetic order in shaping the Fermi surface.

To clarify this point, we have added the following sentence on page 9, line 255: "rather than to increased Co doping. This assignment is supported by the observation that the γ_K band bottom does not shift downward with doping (see Fig. 2), which is inconsistent with a simple band-filling picture. Instead, the weakening of the 3Q order alters the Fermi surface topology, naturally leading to the enlarged γ Fermi pockets."

(5) Reviewer's comment: What is the difference between Fig. 3(h) and Fig. 4(b)? They look very similar and appear to be redundant.

Our response: We thank the reviewer for raising this point. Although Fig. 3(h) and Fig. 4(b) display similar overall trends, they provide complementary perspectives. Specifically, Fig. 3(h) illustrates the redistribution of spectral weight at the Fermi surface between the tip of the γ pocket and the M point, while Fig. 4(b) presents the corresponding redistribution in the band dispersion through the ratio of EDC peak intensities at the same momenta. Taken together, these two figures offer a more complete picture of how the 3Q order-induced band reconstruction affects both the Fermi surface topology and the underlying band dispersions.

Response to Reviewer #3:

In the article “Discovery of Van Hove Singularities: Electronic Fingerprints of 3Q Magnetic Order in a Van der Waals Quantum Magnet” H.-L. Luo et al. report an indirect evidence of a 3Q magnetic order formation in Co-doped 2H-TaS₂.

Single crystals of pure 2H-TaS₂ and Co-doped TaS₂, with doping levels ranging from 0.29 to 0.36, were investigated using angle-resolved photoemission spectroscopy (ARPES), with pure TaS₂ serving as the reference sample. A critical doping level of 0.33 was identified, above which the magnetic order in Co-doped TaS₂ transitions from a 3Q antiferromagnetic (AFM) state (at low temperatures) to a coplanar helical AFM order. ARPES measurements reveal indirect signatures of the 3Q magnetic order, through an appearance of new bands whose dispersion cannot be explained in the absence of a 3Q order.

While I find the study interesting and appreciate the careful and detailed experimental work, I have a few questions and concerns that I would like to see addressed before recommending this manuscript for publication.

(1) Reviewer’s comment: Page 2, lines 37-28: The authors write that “Magnetically intercalated TMD systems mark the emergence of a new frontier: van der Waals magnets”. Does this refer to an emergence of vdW magnets from a historic perspective, as today we have quite a few truly magnetic vdW materials that do not need any intercalation?

Our response: We thank the reviewer for this insightful comment. We agree that our original statement was imprecise. Our intention was to highlight the historical significance of magnetically intercalated TMDs in the early exploration of van der Waals magnetism, rather than to imply that they represent the first or only class of vdW magnets. To avoid confusion, we have revised the sentence on page 2, line 40, to read: “Magnetically intercalated TMDs, as an important class of van der Waals magnets, open a new paradigm in TMD research and provide a versatile platform to investigate the interplay between magnetic order and electronic properties.”

(2) Reviewer’s comment: What is the reason for majority of ARPES measurements to be done at doping of 0.32? This is very close to a critical doping of 0.325 where 3Q order is replaced by helical order.

Our response: We thank the reviewer for raising this important point. We would like to clarify that, among all the ARPES data presented in this manuscript, only Fig. 1 is based exclusively on the sample with doping level $x=0.32$. In all other figures where data from $x=0.32$ are shown, they are always presented together with results from other doping levels as part of a broader systematic study, and are not used in isolation.

In Fig. 1, we chose the $x = 0.32$ sample because it provides particularly clear spectra for illustrating the key differences between $\text{Co}_{1/3}\text{TaS}_2$ and pristine TaS_2 , namely the downward shift of TaS_2 -derived bands and the emergence of Co-derived bands. The use of the $x = 0.32$ sample does not affect these conclusions, as these features remain robust across the entire doping range studied. Therefore, the conclusions drawn from the $x=0.32$ sample in Fig. 1 are valid and representative of the general behavior.

(3) Reviewer's comment: Page 5, lines 128-129: The authors write that the direct comparison of TaS_2 and $\text{Co}_{0.32}\text{TaS}_2$ reveals several striking differences, but only 2 are listed (significant electron-doping of α and β bands and appearance of a δ band). While two is technically several, it would be more correct to explicitly write "two striking differences", except if this refers to also effects (i)-(iv) on page 6.

Our response: We thank the reviewer for the helpful suggestion. To improve clarity, we have modified the wording on page 5, line 136, replacing "several" with "two."

(4) Reviewer's comment: Page 10, growth methods: Are 0.36 samples grown the same way as 0.29, 0.32 and 0.34? 0.36 samples are not listed in the growth methods.

Our response: We thank the reviewer for pointing this out. The omission of $x=0.36$ was unintentional. The $x=0.36$ samples were grown using the same procedure as the other doping levels. We have now corrected the text and explicitly included $x=0.36$ on page 12, line 331 in the Methods section.

(5) Reviewer's comment: On page 10 it says that ARPES measurement temperature was 10 K, however, experimental methods list 7 K. What is the correct measurement temperature?

Our response: We thank the reviewer for this careful observation. The correct ARPES measurement temperature is 7 K. The mention of 10 K on page 4 was a typographical error, which we have now corrected it in the revised manuscript (page 4, line101).

(6) Reviewer's comment: Page 11, ARPES details: What is the reason for performing TaS_2 and Co-doped TaS_2 measurements at different photon energies? Is it related to their matrix elements, or to a different size of the 3D unit cell?

Our response: We thank the reviewer for this question. Different photon energies were employed for TaS_2 and Co-doped TaS_2 because of matrix element effects. In our experiments, we performed photon energy-dependent measurements and selected the photon energy that provided the sharpest and most well-defined band structure for each sample. This approach ensures optimal

contrast and visibility of the relevant electronic features in ARPES spectra.

(7) Reviewer's comment: In the article authors write that there are 2 different terminations one can end up with after cleaving, TaS₂ one and Co one. Am I correct to assume that the data shown in the main article is from the TaS₂ plane? This should be made more clear in the article.

Our response: We thank the reviewer for this suggestion. All the band structure data presented in the main figures were indeed measured on the TaS₂-terminated surface. To clarify this point, we have revised the sentence on page 5, line 115, from "Overall, the data of the TaS₂-termination presented in Fig. 1f show much sharper band structure." to "All band structures presented here are from the TaS₂-terminated surface, as it provides much sharper band structures."

(8) Reviewer's comment: Related to my previous question, do authors observe the same type of behaviour (appearance of the same bands, as well as the same trends with doping) on the Co plane as well?

Our response: We thank the reviewer for raising this question. We show in Supplementary Fig. S1 the band structure measured on a Co-terminated surface. During cleaving, the Co atoms located between TaS₂ layers are partially retained on the exposed surface and partially removed with the opposite side, leading to a disrupted atomic arrangement. As a result, the residual Co atoms do not form a well-ordered lattice, preventing the emergence of well-defined Co-derived bands. In addition, due to the limited photoelectron escape depth, Co-derived states beneath the first TaS₂ layer are not detectable. Since the Co-derived bands are central to our study but are not clearly observed on the Co-terminated surface, we did not pursue doping-dependent measurements for this termination.

(9) Reviewer's comment: Page 9 in the SI, Figure S8: The authors observe a Mexican hat-like dispersion for samples with 0.29, 0.32 and 0.33 doping, while at 0.34 and 0.36 this turns into a simple hole-like band without a dip at the M point. Can authors comment as to why they see the dip also at 0.33 doping since this is above the critical doping value (0.325 listed in the main text) and one would expect that 0.33 should behave the same way as 0.34 and 0.36

Our response: We thank the reviewer for this insightful question. Although the doping level $x = 0.33$ is slightly above the critical point ($x \approx 0.325$) where the 3Q order is expected to be replaced by the helical phase, we still observe a dip at the M point in the ARPES band structure, similar to lower dopings. We attribute this to the crossover nature of the magnetic transition: the change from 3Q to helical order does not occur abruptly at a single doping value, but rather spans a finite doping range.

Furthermore, even though the anomalous Hall effect is no longer observed at $x = 0.33$, the

spectral signature at the M point may persist due to short-range magnetic order or spatial inhomogeneity near the phase boundary. Such effects can produce a coexistence or competition of magnetic orders within a narrow doping window. By contrast, for $x = 0.34$ and 0.36 , the dip is completely absent, consistent with the system having fully entered the helical regime.

To clarify this point, we have added the following sentence to the caption of Supplementary Fig. S8 (now Fig. S11): “For $x = 0.33$, although transport measurements no longer show an anomalous Hall effect, a remnant dip at the M point remains visible in ARPES, which may originate from short-range magnetic order or spatial inhomogeneity near the phase boundary.”

Summary of Modifications:

1. Following Comment #1 of Referee #1, we have added Fig. R1 as Supplementary Fig. S5, and the corresponding *DFT Calculations* section in *Methods*. In addition, we have included a discussion on page 6, line 162: “We further performed DFT calculations for $\text{Co}_{1/3}\text{TaS}_2$ and identified highly-dispersive bands mainly originating from Ta orbitals together with near- E_f flat bands primarily deriving from Co orbitals (see Supplementary Fig. S5).”
2. Following Comment #1 of Referee #1 and Comment #2 of Referee #2, we have added a discussion on page 8, line 219: “Since DFT struggles to accurately describe the dispersion of the near- E_f flat bands, primarily due to strong electron-electron correlations and the presence of local magnetic moments, it cannot provide a reliable account of our system. At the same time, the near- E_f flat bands are dominated by Co $3d$ orbitals, whereas the dispersive α and β bands mainly originate from Ta $5d$ states (see Supplementary Fig. S5). Given both the limitations of DFT and the distinct orbital characters of these states, we adopt a phenomenological tight-binding model on a Co triangular lattice to account for the flat bands and their coupling to local magnetic moments.”
3. Following Comment #1 of Referee #1, we have included Fig. R2 as Supplementary Fig. S12 and added the following sentence on page 10, line 283: “We further tested the robustness of this result by varying hopping amplitude t and coupling constant J , and found that the inverse Mexican hat-like dispersion persists as long as the 3Q order is present (See Supplementary Fig. S12).”
4. Following Comment #2 of Referee #1, we have revised the sentence “namely, when the magnetic moments are randomly oriented” to “that is, in the non-magnetic state (exchange coupling constant $J=0$)” on page 8, line 228.
5. Following Comment #4 of Referee #1 and Comment #2 of Referee #2, we have added Fig. R5 as Supplementary Fig. S14, Fig. R4 as Supplementary Fig. S15, and the corresponding *Theory* section in *Methods*.
6. Following Comment #4 of Referee #1, we have added the discussion on page 11, line 304: “Temperature-dependent measurements provide further evidence for the role of 3Q magnetic order. As shown in Supplementary Fig. S14, the inverse Mexican hat-like dispersion is observed only in the 3Q ordered phase at low temperature. At higher temperatures, corresponding to the 1Q ordered and paramagnetic states, the dispersion evolves into a simple hole-like band along the K-M-K direction. These observations are in close agreement with tight-binding calculations,

which shows that only the 3Q ordered state gives rise to the inverse Mexican hat-like dispersion (Fig. 4 and Supplementary Fig. S15). Together, these results demonstrate that the reconstruction of the near-E_f bands is a distinctive fingerprint of the 3Q order.”

7. Following Comment #7 of Referee #1, we have added Fig. R7 as Supplementary Fig. S4
8. Following Comment #7 of Referee #1, we have added a sentence on page 6, line 149: “Density Functional Theory (DFT) calculations for 2H-TaS₂ reveal that the observed ~300meV band shift can be reproduced by adding one electron per TaS₂ unit cell, in good agreement with a simple electron-counting picture of Co_{1/3}TaS₂ consisting of alternating [Co_{1/3}]⁺ and [TaS₂] layers (see Supplementary Fig. S4).”
9. Following Comment #7 of Referee #1, we have added a sentence on page 7, line 189: “The observed slope of the shift of the α -band minimum shift as a function of Co doping is well reproduced by a rigid-band-shift approximation applied to the TaS₂ band structure (see Supplementary Fig. S4).”
10. Following Comment #8 of Referee #1, we have added the phrase “a pronounced reduction of spectral weight in the highly-dispersive α and β bands” on page 6, line 170.
11. Following Comment #8 of Referee #1, we have added “and the DFT-calculated orbital projected band structures (see Supplementary Note and Supplementary Fig. S6-S7)” on page 6, line 172.
12. Following Comment #8 of Referee #1, we have revised the sentence on page 7, line 174, to read: “the near-E_f γ_K and γ_M bands mainly originate from Co 3d_{z²} orbital, the highly-dispersive α and β bands are dominated by Ta 5d_{z²} Ta 5d_{xy/dx²-y²} orbitals, and the ϵ band is primarily contributed by d_{xz/dyz} and d_{xy/dx²-y²} states.”
13. Following Comment #8 of Referee #1, we have added Fig. R8 as Supplementary Fig. S6, Fig. R9 as Supplementary Fig. S7, and included the corresponding discussion as a Supplementary Note in the Supplementary Materials.
14. Following Comment #3 of Referee #2, we have renamed the current γ band as γ_K (along Γ -K) and the previous δ band as γ_M (Γ -M). In addition, the previous labels for the ϵ and ζ bands are now labeled δ and ϵ .
15. Following Comment #4 of Referee #2, we have added the following sentences on page 9, line 255: “rather than to increased Co doping. This assignment is supported by the observation that the γ_K band bottom does not shift downward with doping (see Fig. 2), which is inconsistent with a simple band-filling picture. Instead, the weakening of the 3Q order alters the Fermi surface topology, naturally leading to the enlarged γ Fermi pockets.”
16. Following Comment #1 of Referee #3, we have revised the sentence on page 2, line 40, to read: “Magnetically intercalated TMDs, as an important class of van der Waals magnets, open a new paradigm in TMD research and provide a versatile platform to investigate the interplay between magnetic order and electronic properties.”
17. Following Comment #3 of Referee #3, we have revised the wording on page 5, line 136, changing “several” to “two”.
18. Following Comment #4 of Referee #3, we have included “x=0.36” on page 12, line 331 in the *Methods* section.
19. Following Comment #5 of Referee #3, we have revised “10K” to “7K” on page 4, line 101.
20. Following Comment #7 of Referee #3, we have revised the sentence on page 5, line 115, to read: “All band structures presented here are from the TaS₂-terminated surface, as it provides much sharper band structures.”

21. Following Comment #9 of Referee #3, we have added the following sentence to the caption of Supplementary Fig. S11: “For $x = 0.33$, although transport measurements no longer show an anomalous Hall effect, a remnant dip at the M point remains visible in ARPES, which may originate from short-range magnetic order or spatial inhomogeneity near the phase boundary.”
22. We have moved Debasis Dutta and Guoqing Chang from the Acknowledgements to the list of co-authors, as they contributed to the DFT calculations. Their funding information has been added to the *Acknowledgments*.

References:

1. P. Park, et al. Field-tunable toroidal moment and anomalous Hall effect in noncollinear antiferromagnetic Weyl semimetal $\text{Co}_{1/3}\text{TaS}_2$. *npj Quantum Materials* **7**, 42 (2022).
2. P. Park, et al. Tetrahedral triple-Q magnetic ordering and large spontaneous Hall conductivity in the metallic triangular antiferromagnet $\text{Co}_{1/3}\text{TaS}_2$. *Nature Communications* **14**, 8346 (2023)
3. L. Xie, et al. Structure and Magnetism of Iron- and Chromium-Intercalated Niobium and Tantalum Disulfides. *Journal of the American Chemical Society* **144**, 9525-9542 (2022).
4. L. Xie, et al. Comparative Electronic Structures of the Chiral Helimagnets $\text{Cr}_{1/3}\text{NbS}_2$ and $\text{Cr}_{1/3}\text{TaS}_2$. *Chemistry of Materials* **35**, 7239-7251 (2023).
5. S. Parkin, et al. 3d Transition-Metal Intercalates of the Niobium and Tantalum Dichalcogenides. I. Magnetic Properties. *Philosophical Magazine B* **41**, 65-93 (1980).
6. A. Damascelli, et al. Angle-Resolved Photoemission Studies of the Cuprate Superconductors. *Reviews of Modern Physics* **75**, 473-541 (2003).

Response to Reviewer's Comments:

Response to Reviewer #1:

I went through the authors' response letter, and I think they addressed my questions/concerns properly. Therefore, I can recommend the publication of this manuscript.

We thank the referee for carefully reading our revised manuscript and response letter and for recommending publication.

Response to Reviewer #2:

I thank the authors for the efforts to answer all the relevant questions. Although I find that most of issues are well addressed, I am still not convinced that the new data in Fig. S14 provides "additional" support for the electronic signature of the 3Q magnetic phase. Specifically, the temperature-dependent change in the EDC of Fig. S14(d-f) is very subtle and can be easily affected by the thermal broadening, i.e., the Mexican-hat bands might still be present in the non-magnetic phase, but it becomes invisible due to thermal broadening at elevated temperatures. In general, the temperature-induced change in the experimental band structure is very small compared to theoretical predictions. Some clarification on this point will certainly be important to a comprehensive understanding of the possible 3Q magnetic phase.

We thank the referee for this thoughtful comment. To clarify whether the Mexican hat-like dispersion can be obscured by thermal broadening at 82K, we performed quantitative simulations that incorporate the sources of thermal broadening: the Fermi-Dirac cutoff and the temperature evolution of the intrinsic scattering rate.

To model the ARPES spectra, we start from an intrinsic spectral peak represented by a Lorentzian $L(E)$ centered at $E - E_F = E_0$ with a full width at half maximum (FWHM) Γ . The temperature-dependent occupation is introduced by multiplying this spectrum by the Fermi-Dirac distribution $f(E, T)$. The resulting curve is then convolved with the experimental energy resolution, which is modeled as a Gaussian $G(E; \Delta E_{res})$ with FWHM ΔE_{res} . The simulated energy distribution curve (EDC) is therefore

$$I_{sim}(E, T) = [L(E)f(E, T)] \otimes G(E),$$

where

$$L(E) = \frac{1}{\pi} \frac{\Gamma/2}{(E-E_0)^2 + (\Gamma/2)^2}, \quad f(E, T) = \frac{1}{\exp(\frac{E-E_F}{k_B T}) + 1},$$

and

$$G(E; \Delta E_{res}) = \frac{1}{\sigma\sqrt{2\pi}} \exp(-\frac{E^2}{2\sigma^2}), \quad \sigma = \frac{\Delta E_{res}}{2\sqrt{2} \ln 2},$$

with k_B the Boltzmann constant and \otimes denoting convolution [1].

To simulate the 6K spectra shown in Fig. R1g, we set $T=6\text{K}$, $\Delta E_{res} \approx \sqrt{\Delta E_{analyzer}^2 + \Delta E_{hv}^2} = 10.6\text{meV}$, and $\Gamma=24\text{meV}$, and choose $E_0 = -6\text{meV}$ and -2meV to best reproduce experimental EDC(k1) and EDC(k2) from Fig. R1d, respectively. The inverse Mexican hat-like dispersion is reflected in the higher binding energy of the EDC(-6meV) peak relative to the EDC(-2meV) peak. To examine whether an increased temperature could smear out the Mexican-hat feature, we then simulate the 31K and 82K spectra (Fig. R1h and R1i). In these simulations, the peak-center energies E_0 are kept fixed at -6meV and -2meV for the orange and blue curves, respectively, and two physically motivated quantities are varied: (i) the temperature in the Fermi-Dirac distribution and (ii) the Lorentzian FWHM (Γ), which is adjusted to best reproduce the experimental data. The latter is allowed to increase modestly with temperature because the intrinsic linewidth reflects the imaginary part of the self-energy (i.e., the intrinsic scattering rate), which generally grows with temperature due to electron-phonon and electron-electron interactions.

Importantly, these simulations show that over the temperature range studied (6-82 K), even after including both sources of thermal broadening, an inverse Mexican hat-like dispersion that persists to high temperature would still produce a clearly deeper EDC peak for $E_0 = -6\text{meV}$ than for $E_0 = -2\text{meV}$. Thermal broadening alone therefore cannot hide such an inverse Mexican hat-like dispersion within our experimental temperature window. In contrast, the 82K data (Fig. R1f) exhibit the opposite behavior: the EDC(k1) peak becomes slightly shallower than the EDC(k2) peak. Thus, the comparison between panels (d-f) and (g-i) shows that the hole-like dispersion observed at 31K and 82K cannot be generated by thermal broadening of an inverse Mexican hat-like band and is instead intrinsic. Notably, stronger thermal broadening could in principle smear out this characteristic inverse Mexican hat-like band. As illustrated by the example in the inset of Fig. R1i, when the temperature is raised to 200 K and the FWHM is increased to $\Gamma=100\text{meV}$, the two EDCs nearly coincide, such that the underlying inverse Mexican hat-like band becomes effectively indiscernible.

In the revised Supplementary Materials, we have incorporated Fig. R1g-R1i into Supplementary Fig. S14, added Supplementary Note 2 to explain the temperature-dependent data, and included the above discussion as Supplementary Note 3 to clarify that the hole-like dispersion at 31K and 82K is intrinsic rather than a thermally masked inverse Mexican hat-like dispersion. In the revised main text (page 11), we explicitly refer to Supplementary Notes 2 and 3 when discussing the temperature-dependent measurements.

Fig. R1. Temperature-dependent ARPES spectra of $\text{Co}_{0.31}\text{TaS}_2$ along the K-M-K direction. **(a-c)** Band structures measured at 6K, 31K, and 82K, respectively. Arrows mark the momenta k_1 (M point) and k_2 . **(d-f)** EDCs extracted at k_1 and k_2 from panels (a-c). **(g-i)** Simulated EDCs at 6 K, 31 K, 82 K, and 200K (inset of panel i), assuming that the underlying band dispersion remains the same inverse Mexican hat-like dispersion as at 6 K, i.e., all orange and blue curves generated from a Lorentzian lineshape $L(E)$ centered at $E_0 = -6$ meV and -2 meV, respectively. The FWHM value Γ used for simulations at 6 K, 31 K, and 82 K are chosen to best reproduce the experimental EDCs in panels (d-f).

Reference:

1. A. Damascelli, et al. Angle-Resolved Photoemission Studies of the Cuprate Superconductors. *Reviews of Modern Physics* **75**, 473-541 (2003).

Response to Reviewer' s Comments:

Response to Reviewer #2:

I thank the authors for addressing this question. The detailed simulation now provides a compelling evidence for the electronic transition associated with the triple-Q phase. Therefore, I recommend the publication of this paper in Nature Communications in the current form.

We thank the referee for carefully reading our revised manuscript and response letter and for recommending publication.